# High Frequency radar-derived coastal upwelling index

Pablo Lorente[1], Anna Rubio[2], Emma Reyes[3], Lohitzune Solabarrieta[2], Silvia Piedracoba[4], Joaquín Tintoré[3,5], Julien Mader[2]

[1]Puertos del Estado, Madrid, 28042, Spain
[2]AZTI Marine Research, Basque Research and Technology Alliance (BRTA), Pasaia, 20110, Spain
[3]Balearic Islands Coastal Ocean Observing and Forecasting System (SOCIB), Palma, 07122, Spain
[4]CETMAR (Centro Tecnológico del Mar), Vigo, 36208, Spain
[5]Mediterranean Institute for Advanced Studies (IMEDEA), Esporles, 07190, Spain

*Correspondence to*: Pablo Lorente (plorente@puertos.es)

**Abstract.** Coastal upwelling has been extensively studied since it plays a critical role in the connectivity between offshore waters and coastal ecosystems, which has impacts on water quality, fisheries and aquaculture production. Significant efforts have been devoted to the quantification of the intensity, duration and variability of this phenomenon by means of coastal upwelling indexes (CUI), derived from wind, sea level pressure or sea surface temperature data. Albeit valuable first-order descriptors, such classical indexes have been reported to present some limitations. As one of the major shortcomings is the omission of the direct influence of ocean circulation, this work introduces a novel CUI, generated from remote-sensed hourly surface current observations provided by a High-Frequency radar (HFR). The consistency of the proposed index (CUI-HFR) is assessed in two different oceanographic areas during two distinct time periods: in the North-western Iberian (NWI) Peninsula for 2021 and in the Bay of Biscay (BOB) for 2014, respectively. To this aim, CUI-HFR is compared against a traditional CUI based on hourly wind observations (CUI-WIND) provided by two buoys. Likewise, the skill of CUI-HFR to identify upwelling and downwelling processes is also qualitatively evaluated. Complementarily, the prognostic capabilities of the GLOBAL analysis and forecasting system to accurately reproduce upwelling and downwelling events in the NWI area are also analysed (CUI-GLOBAL). Results obtained in these two pilot areas revealed: (i) a noticeable agreement between CUI-HFR and CUI-WIND, with correlation coefficients above 0.67; (ii) a proven ability of CUI-HFR and CUI-GLOBAL to categorize a variety of upwelling and downwelling episodes, which highlights their potential applicability for direct upwelling monitoring over any coastal area of the global ocean.

## 1 Introduction

Since coastal waters encompass many unique habitats and support a wide range of anthropogenic activities (tourism, transportation, fisheries, etc.), its accurate monitoring is required for an efficient marine resources management, the preservation of vulnerable ecosystems and the sustained development of the so-called blue economy (Trincardi et al., 2020). A cutting-edge technology that has been steadily gaining worldwide recognition as an effective shore-based remote sensing instrument is high-frequency radar -HFR- (Roarty et al., 2019). HFR networks have become an essential component of ocean

observatories as they collect, in near-real time, fine-resolution maps of the upper-layer flow over broad coastal areas, providing a dynamical framework for other traditional in situ observation platforms (Lorente et al., 2022; Rubio et al., 2017). HFR-derived surface circulation is a reliable source of information for search-and-rescue operations and oil spill tracking,

among other practical applications (Reyes et al., 2022; Roarty et al., 2019; Rubio et al. 2017). Equally, it can be used for a detailed investigation of upwelling (UPW) and downwelling (DOW) processes that modulate the connectivity between offshore waters and coastal ecosystems (Lorente et al., 2020; Paduan et al., 2018; Kaplan and Largier, 2006). It occurs when along-shore winds and the Coriolis effect (due to Earth's rotation) combine to drive a near-surface layer of water offshore, a process referred to as Ekman transport (Ekman, 1905). Such cross-shelf transport is compensated by the vertical uplift of

cold and enriched waters that fertilize the uppermost layer. Conversely, during DOW events winds induce a net onshore displacement and subduction of surface coastal waters that foster the retention of organic matter and pollutants onto the shoreline with the subsequent impact on residence times and water renewal mechanisms.

Nowadays, there is an emerging question about the impact of anthropogenic pressures on future changes in coastal UPW ecosystems and the potential implications for biodiversity conservation (IPCC 2022; Abrahams et al., 2021a; Xiu et al.,

2018; Bakun et al., 2015; Cropper et al., 2014; Di Lorenzo, 2015). It has been hypothesized that climate change could promote greater land-sea temperature gradients together with stronger alongshore winds, eventually resulting in intensified wind-driven coastal UPW (Di Lorenzo, 2015; Varela et al., 2015; Barton et al., 2013). In Wang et al. (2015), results from an ensemble of climate models showed relevant changes in the timing, duration, intensity, and spatial heterogeneity of coastal UPW in response to future warming in most Eastern Boundary Upwelling Systems.

Since there is no available ground-truth observation of vertical currents, diverse coastal upwelling indexes (CUI) have been derived from historical estimations of other met-ocean parameters (e.g., wind, sea level pressure or sea surface temperature) to indirectly infer seasonal trends of UPW along the shoreline (Gunduz et al., 2022; Mourre et al., under review; Abrahams et al., 2021a and 2021b; González-Nuevo et al., 2014; Benazzouz et al., 2014; Wooster et al., 1976; Bakun, 1973). Albeit valuable first-order descriptors, such classical indexes have been reported to provide an incomplete picture of coastal UPW

due to a number of limitations, encompassing uncertainties related to the estimation of wind stress or derived from the coarse spatiotemporal resolution of the atmospheric pressure fields used (Jacox et al., 2018).

As one of the major shortcomings is the omission of the direct influence of ocean circulation, the present work intends to fill this gap by introducing a novel CUI, generated from HFR-derived hourly surface current observations (CUI-HFR). Although such ocean-based index was firstly presented in Lorente et al. (2020) as a proof-of-concept investigation, here we gain

further insight into the credibility of the proposed approach by analysing two different regions (Figure 1, a), the North-Western Iberian (NWI) Peninsula Upwelling system and the Bay of Biscay (BOB), for distinct time periods (2021 and 2014, respectively).

In the NWI region, which extends from 40°N to 44°N (Figure 1, a), the ocean dynamics is regulated by the relative changes in the strength and position of the Azores high-pressure system with respect to the Iceland low-pressure system, defining two

largely wind-driven oceanographic seasons (Lorente et al., 2020). The UPW season is predominant from March to October

when northerly winds prevail and induce a south-westward surface flow. The upward flux of cold deep waters supplies nutrients to the euphotic zone, resulting in an elongated along-shore strip of maximum concentration of chlorophyll (Figure 1, b). During the rest of the year, the predominant DOW season is characterized by i) prominent southerly winds that drive the surface circulation northwards; ii) low biological productivity, with nutrients concentration confined in shallower coastal areas (Figure 1, c).

The south-eastern BOB (Figure 1, a) is characterized by a complex topography where a combination of different temporal and spatial ocean processes (i.e., slope currents, eddies, etc.) govern the local hydrodynamics (Llope et al., 2008). Coastal UPW and DOW processes depict a marked seasonal variability, linked to seasonal changes of the wind regime (Prego et al., 2012; Botas et al., 1990). During autumn and winter, south-westerly winds are prevalent and induce DOW events. In spring and summer, winds are less intense and more variable, with dominance of easterlies and north-easterlies driving weak but frequent UPW episodes (Fontán et al., 2008). Although UPW (whenever present) can be considered weak in comparison with typical UPW areas worldwide (Borja et al., 2008), this phenomenon has been investigated in BOB region as they are significantly correlated with the recruitment of commercial fish species (Caballero et al., 2018). In particular, the 55% of the recruitment variability of anchovy fishery can be explained by upwelling over the spawning area (Borja et al., 2008). Therefore, the HFR deployed in this region might be very useful to monitor the surface circulation and characterize the UPW variability by means of a CUI (Caballero et al., 2018).

The main goal of this contribution is twofold. Firstly, to assess the validity of the proposed ocean-based CUI-HFR through its comparison against a traditional CUI based on hourly wind observations (CUI-WIND) provided by adjacent in situ buoys (Figure 1, a). Likewise, the skill of CUI-HFR to categorize UPW and DOW events was also qualitatively evaluated. Secondly, to infer the prognostic capabilities of the GLOBAL analysis and forecasting system (Lellouche et al., 2018) to accurately reproduce UPW and DOW events previously detected in the NWI region. In this context, hydrodynamic models can serve as ancillary tools for gappy HFR surface current maps by offering a seamless predictive picture of the ocean state. This work is structured as follows: Section 2 outlines the observational and modelled data sources. Section 3 describes the methodology adopted. Results are presented and discussed in Section 4. Finally, principal conclusions are drawn in Section 5.

## 2 Data

All the observational and modelled products used in this study are gathered in Table 1 and thoroughly described below.

### 2.1 The HFR system deployed in NWI

A five-site CODAR SeaSonde HFR network, deployed along the Galician-Portuguese coast (Figure 1, b) since 2004, was used in this work (product ref. no. 1 and no. 2 in Table 1). The network is jointly operated by Puertos del Estado, INTECMAR–Xunta de Galicia and the Portuguese Hydrographic Institute. While the southernmost HFR site operates at 13.5

MHz, the other four remaining sites operate at a central frequency of 4.86 MHz, providing hourly radial vectors that are representative of the currents moving toward or away from the site. All those radial current vectors (from two or several sites) within a predefined search radius of 25 km are geometrically combined to estimate hourly total current vectors on a Cartesian regular mesh of 6×6 km horizontal resolution and 200 km range (Lorente et al., 2015 and 2020). Data are quality-controlled and delivered freely through the European HFR Node, which oversees the harvesting, harmonization, formatting, and distribution of HFR data (Corgnati et al., 2021).

HFR-derived data used in this study were collected from August to December 2021, once the southernmost site (LEÇA) was calibrated and integrated in the pre-existing network. During this 5-month period, the five sites were simultaneously operational, and the spatial coverage was at its maximum extent. The specific geometry of the HFR domain and, hence, the intersection angles of radial vectors handicap the accuracy of the total current vectors resolved at each grid point. Such a source of uncertainty is quantified by a dimensionless parameter denominated Geometrical Dilution of Precision -GDOP- (Chapman et al., 1997; Barrick, 2006), which typically increases with the distance from the HFR sites. In this work, a cut-off filter of 3 was imposed for the GDOP, to get rid of those estimations affected by higher uncertainties (Annex 1, a).

## 2.2 The HFR system deployed in BOB

This HFR system, deployed in the South-East BOB since 2009 and owned by Euskalmet (product ref. no. 1 and no. 2 in Table 1), is a two-site CODAR SeaSonde network (Figure 1, a) that operates at a central frequency of 4.46 MHz, providing hourly current maps (representative of the first 1.5 m of the water column) with a spatial resolution of 5 km in an area up to 150 km from the coast (Solabarrieta et al., 2014). The consistent performance of this system and its potential for the study of ocean processes and transport patterns have already been demonstrated by previous works (Manso-Narvarte et al., 2018; Solabarrieta et al., 2014, 2015 and 2016; Rubio et al., 2011 and 2018).

Data are collected, processed and quality-controlled following standard recommendations (Mantovani et al., 2020) and later distributed in near real time through the European HFR Node, as part of the Copernicus Marine Service products. Again, a cut-off filter of 3 was imposed for the GDOP parameter to mitigate the impact of geometrical uncertainties on total current vectors accuracy (Annex 1, b). Finally, HFR-derived current estimations used in this work were collected from April to August 2014, when the system operated consistently and provided the longest data series (Solabarrieta et al., 2015).

## 2.3 In situ buoys

Basic features of two deep ocean buoys (product ref. no. 3 in Table 1), deployed within each HFR footprint (Figure 1, a) and operated by Puertos del Estado, are gathered in Table 2. Both in situ devices collect quality-controlled estimations of sea surface temperature, salinity, and currents, among other physical parameters. Furthermore, both buoys are equipped with a wind sensor, providing hourly averaged wind vectors at 3 m height.

## 2.4 GLOBAL analysis and forecasting system

The operational Mercator global ocean analysis and forecast system provides 10 days of 3D global ocean forecasts updated daily. This product (product ref. no. 4 in Table 1), named GLOBAL_ANALYSIS_FORECAST_PHY_001_024 and freely available through the Copernicus Marine Service catalogue, includes daily and monthly mean files of temperature, salinity, currents, sea level, mixed layer depth, and ice parameters from the surface to seafloor (5500 m depth) over the global ocean. It also includes hourly mean surface fields for sea level height, temperature, and currents. The global ocean output files are displayed on 50 vertical levels with a 1/12° horizontal resolution over a regular latitude and longitude equirectangular projection. This product presents a temporal coverage from November 2020 to present.

The system is based on the Nucleus for European Modelling of the Ocean (NEMO) v3.1 ocean model (Madec, 2008) and is forced with 3-hourly atmospheric fields provided by the European Centre for Medium-Range Weather Forecasts. The GLOBAL system does not include tides nor pressure forcing. Altimeter data, in situ temperature and salinity vertical profiles, and satellite sea surface temperature data are jointly assimilated to estimate the initial conditions for numerical ocean forecasting. For further details, the reader is referred to the GLOBAL Product User Manual (Le Galloudec et al., 2022) and to Lellouche et al. (2018).

## 2.5 Satellite-derived data

Surface fields of temperature and chlorophyll (product ref. no. 5 and no. 6 in Table 1, respectively) provided by satellite missions were downloaded from the Copernicus Marine Service catalogue and used to analyse the impact of UPW and DOW events in their spatial distribution.

## 3 Methodology

Bakun (1973, 1975) proposed a CUI (the so-called Bakun Index) based on Ekman theory (Ekman, 1905) and available estimates of atmospheric conditions to derive estimates of cross-shore Ekman transport as a proxy for coastal UPW. Following the same approach, in the present contribution two CUIs were computed (as benchmarks to compare CUI-HFR against) from hourly wind estimations provided by those in situ buoys moored within each HFR coverage (Figure 1, a). Then, the CUI-WIND is defined as follows:

For the NWI region:
$$U_{Ekman} = CUI - WIND\left(\frac{m^3}{s \cdot km}\right) = -\frac{\rho_a \cdot C_d \cdot \sqrt{u^2+v^2} \cdot v \cdot 1000}{f \cdot \rho_w} \qquad [1]$$

For the BOB region:
$$V_{Ekman} = CUI - WIND\left(\frac{m^3}{s \cdot km}\right) = -\frac{\rho_a \cdot C_d \cdot \sqrt{u^2+v^2} \cdot u \cdot 1000}{f \cdot \rho_w} \qquad [2]$$

where $\rho_a$ is the air density at standard temperature and pressure conditions (1.22 kg·m$^{-3}$), $\rho_w$ is the seawater density (1025 kg·m$^{-3}$), $C_d$ is a dimensionless empirical Drag coefficient (1.4·10$^{-3}$) and $f$ is the latitude-dependent Coriolis parameter (whose values are shown in Table 2). In this case, u and v denote the hourly time series of zonal and meridional wind at 3 m

height, respectively, as measured by each buoy. The sign is changed to define positive (negative) magnitudes of CUI-WIND as response to the predominant equatorward (poleward) wind over NWI and the predominant westwards (eastwards) wind over BOB.

Assuming the prompt and direct reaction of the upper ocean layer to intense and prolonged wind forcing in NWI (Herrera et al., 2005) and BOB (Solabarrieta et al., 2015), it seems reasonable to develop an ocean-based indicator for UPW and DOW conditions. Analogously, the CUI-HFR is defined as follows:

For the NWI region:
$$U_{Ekman} = CUI - HFR\left(\frac{m^3}{s \cdot km}\right) = -\frac{\rho_a \cdot C_d \cdot \sqrt{u^2 + v^2} \cdot v \cdot 1000}{f \cdot \rho_w} \cdot C \qquad [3]$$

For the BOB region:
$$V_{Ekman} = CUI - HFR\left(\frac{m^3}{s \cdot km}\right) = -\frac{\rho_a \cdot C_d \cdot \sqrt{u^2 + v^2} \cdot u \cdot 1000}{f \cdot \rho_w} \cdot C \qquad [4]$$

where u and v represent the filtered hourly time series of zonal and meridional surface current velocities (m·s$^{-1}$) provided by the HFR, respectively. A 25-h running-mean filter was used to smooth time-series data by suppressing the main diurnal and semidiurnal tidal constituents (Shirakata et al., 2016), particularly the M2 signal, which is the largest harmonic constituent in the study area. Furthermore, C represents a dimensionless parameter that acts as proportionality constant between wind and surface current observations, assuming the direct relationship between both parameters (Lorente et al., 2020; Solabarrieta et al., 2015). The value of this parameter, which depends on the selected study area, can be easily derived from the best linear fit against CUI-WIND. For instance, C is equal to 2300 (3500) for the NWI (BOB) region.

In order to obtain a single time series of CUI-HFR, which is representative of the entire study area, hourly 2D maps were spatially averaged over a delimited HFR spatial subdomain (see Annex 1). In particular, two prerequisites were imposed to select such subdomain: i) it should be comprised between 60 and 1200 m bathymetric depths to represent coastal waters while avoiding shallow water effects and the potential impact of impulsive-type freshwater discharges on the coastal circulation; ii) A predefined threshold of 3 was established for the GDOP parameter to minimize geometrical uncertainties in the remote-sensed currents. Finally, as a rough approximation, we also assumed that the local coastline was perfectly aligned in the South-North axis for the NWI region and in the East-West axis for the BOB region, which implies that v and u surface current components were parallel to the coastline, respectively.

Representative UPW and DOW episodes were selected for each study area to analyse the prevailing met-ocean conditions (wind and surface circulation, among others) along with the spatial distribution of CUI. In the case of NWI area, a CUI was also derived from the surface currents predicted by the GLOBAL analysis and forecast system (Lellouche et al., 2018), following equation [3]. CUI-GLOBAL and CUI-HFR were intercompared to infer strengths and weaknesses of each index and infer the potential of GLOBAL model as predictive tool for UPW conditions in this specific region. To this aim, a variety of outcomes were computed, ranging from 2D maps, time series and best linear fit of scatter plots to Hovmöller diagrams, which are the common way of plotting met-ocean data to depict changes over time of scalar quantities such as the CUI (Benazzouz et al., 2014).

## 4 Results

To quantify the consistency of both CUI-HFR and CUI-GLOBAL in the NWI area, hourly time series were compared against CUI-WIND, derived from wind estimations provided by Silleiro buoy (Figure 2, a-b). The visual resemblance between the three different CUIs was noticeable, with significantly high correlation coefficients: i) 0.72 and 0.74 between CUI-GLOBAL and CUI-WIND for the entire 2021 and for August-December 2021, respectively, as reflected by their best linear fit of scatter plots (Annex 2, a-b); ii) 0.80 between CUI-HFR and CUI-WIND for August-December 2021 (Annex 2, c); iii) 0.91 between CUI-GLOBAL and CUI-HFR for August-December 2021 (Annex 2, d). According to the statistical results exposed in Annex 2 (a-c), we can state: i) the slope and intercept values were close to 1 and moderately low, respectively; and ii) ocean-based CUI and CUI-WIND are strongly correlated, likely due to the role of alongshore wind stress as primary driver of UPW conditions in the NWI area. Hourly alongshore winds (from Silleiro buoy) and HFR-derived alongshore currents (at the grid point closest to the buoy) are highly correlated (0.80) for August-December 2021 (not shown). It is also worth highlighting the ability of GLOBAL model to capture many of the most relevant UPW (positive values) and DOW (negative values) episodes such as those observed during February 2021, whose values were higher than 3000 $m^3 \cdot s^{-1} \cdot km^{-1}$ and lower than -5000 $m^3 \cdot s^{-1} \cdot km^{-1}$, respectively (Figure 2, a).

With regards to the 5-month period (August-December 2021) of concurrent CUI estimations (Figure 2, b), the degree of agreement was rather satisfactory. Both ocean-based CUIs were able to reproduce the temporal variability, capturing fairly well the most prominent events (marked and denoted in black). Hovmöller diagrams of CUI were computed at a selected transect of constant longitude (9.43°W, shown in Figure 1-a) to infer its spatio-temporal evolution (Figure 2, c-d). There were clear similarities (in terms of intensity and timing) between CUI-HFR and CUI-GLOBAL during the entire 5-month period, with a predominance of UPW (DOW) events during the summer and autumn (early winter) of 2021.

The Hovmöller diagrams of sea surface temperature (SST) and chlorophyll (CHL) concentration at the same transect corroborated the consistency of the proposed approach (Figure 2, e-f). An abrupt cooling (with SST below 14°C) and a relevant peak of CHL (with values above 2 $mg \cdot m^{-3}$) were observed by mid-August 2021, coincident with the UPW-1 event previously categorized (Figure 2, b-d). Equally, the moderate UPW episode that took place the 20[th] of September 2021 (denoted as UPW-2 in Figure 2-b) could be also related to a local drop of SST (Figure 2, e) and a maximum of CHL (Figure 2, f) at the northernmost sector of the transect, in the vicinity of Cape Finisterre (Figure 1, a), which has been largely documented to act as the locus of frequent UPW (Lorente et al., 2020; Torres et al., 2003). Finally, the intense DOW event identified by CUI-HFR and CUI-GLOBAL for the 21[st] of December 2021 (denoted as DOW-1 in Figure 2-b) was coincident with a general minimum of CHL concentration (Figure 2, f), resulting in reduced primary production. These findings confirm the strong connection between the wind-induced circulation, coastal UPW and the modulation of SST and CHL fields at the uppermost layer, in line with previous works. For instance, Alvarez et al. (2012) postulated that the high seasonal variability of CHL in NWI was mainly related to UPW episodes during spring and summer, while CHL variations depended on other additional factors (such as the input of nutrients from land run-off) during autumn and winter.

To gain further insight into the three specific events previously identified (Figure 2, b), daily-averaged maps of CUI and circulation patterns were computed from current estimations provided by both HFR and GLOBAL model (Figure 3). During the UPW-1 (Figure 3, a-b) and UPW-2 (Figure 3, c-d) events, intense northerlies (with gusts above 10 m·s⁻¹) were predominant over the study area, according to the wind roses derived from Silleiro buoy observations. Despite of the observed discrepancies in magnitude and direction, maps of wind-induced surface currents shared some common features: i) the prevailing S-SW surface circulation (as response to northerly winds) along with the typical offshore deflection of the flow, associated with UPW-favourable conditions; ii) the rather uniform circulation westwards (south-westwards) to the north (south) of Cape Finisterre (indicated in Figure 1, a); iii) the absence of submesoscale structures (i.e. eddies, small meanders, etc.) due to the strong wind-induced homogenization. It is also worth mentioning that GLOBAL model predicted a more pronounced along-shore circulation (i.e., a more intense meridional -v- velocity component) to the south of this coastal promontory, which probably gave rise to higher CUI values (Figure 3, b) than those derived from HFR current estimations (Figure 3, a), according to equation [3]. Indeed, this could be also observed in Figure 2-b where the peak of CUI-GLOBAL (green line) was higher than the concomitant peak of CUI-HFR (red line) during the UPW-1 episode. By contrast, local maximums of CUI-GLOBAL and CUI-HFR time series during the UPW-2 event were rather alike in terms of timing and strength (Figure 2, b). Analogously, the spatial distribution of CUI was quite similar for HFR (Figure 3, c) and GLOBAL (Figure 3, d), with a core of maximum UPW observed to the south of Rías Baixas (several coastal embayments characterized by a high biodiversity, denoted in Figure 1-a). The main discrepancy between both patterns was that in the case of GLOBAL, the peak appeared in the form of an elongated belt of positive values of CUI (confined landward close to the shoreline), whereas the HFR-derived pattern revealed a wider dipole-like structure, with two main cores of maximum CUI-HFR (close to 4000 m³·s⁻¹·km⁻¹). The drop of CUI-HFR is consistent (in timing and location) with the drop of CHL concentration showed in Figure 1-b, supporting this HFR pattern which was already documented in Lorente et al. (2020). Finally, during the DOW-1 event (Figure 2, b), both HFR and GLOBAL captured fairly well the so-called Iberian Poleward Current, a narrow surface poleward flow along the NWI shelf edge (Torres and Barton, 2006). As reflected in Figure 3 (e-f), DOW-favourable southerly winds induced a net flow to the north that settled in the continental shelf and circuited the western and northern Iberia margins. Again, the main difference resided in the existence of both a longitudinal gradient and a narrow strip of very negative CUI values close to the coastline in the case of GLOBAL (Figure 3, f), while the HFR-derived map of CUI exhibited a dipole-like distribution where the values were not so negative (Figure 3, e). This was also evidenced in the spatially-averaged hourly time series of CUI shown in Figure 2-b, where the drop in CUI-GLOBAL was sharper than the observed in CUI-HFR.

To further verify the validity of the proposed methodology, a similar CUI was generated for a 5-month period (April-August 2014) from hourly surface current estimation provided by another HFR system deployed in the BOB since 2009 (Solabarrieta et al., 2014). This CUI-HFR was validated against a CUI-WIND based on hourly estimations provided by Bilbao buoy (located within the HFR footprint and depicted in Figure 1-a), used here as a reference benchmark (Figure 4, a). The visual resemblance between both time series was significantly high, with a correlation coefficient and a slope of 0.68

and 1, respectively, for a set of hourly 3672 data (Annex 2, e). During the analysed period, DOW events (5) were predominant with respect to UPW episodes (1). It seemed that CUI-HFR tended to slightly underestimate the intensity of some DOW events (i.e., DOW-2, DOW-3 and DOW-4), while it clearly overestimated the strength of UPW-1 episode (Figure 4, a).

A Hovmöller diagram of CUI-HFR was computed at a selected transect of constant latitude (43.54ºN, denoted in Figure 1-a) to deduce its spatio-temporal variability (Figure 4, b). The 6 events previously categorized were clearly observed and marked in the diagram, where the UPW/DOW features were evidenced throughout the entire transect. A number of secondary UPW events could be observed: albeit relevant in strength and duration, they were spatially confined to the westernmost sector of the transect (coincident with Cape Matxitxako, denoted in Figure 1-a) during the beginning of April, late July, and early August 2014. This fact highlighted the importance of coastal promontories as modulators of UPW processes by inducing important wind stress variations and zones of retention (Pitcher et al., 2010).

During DOW-1 event (Figure 4, c), intense westerly winds (up to 10-12 m·s$^{-1}$) were dominant and a counter-clockwise recirculation pattern prevailed, revealing the associated mechanisms of convergence and subduction (and the subsequent vertical mixing), especially in the vicinity of Cape Matxitxako (denoted in Figure 1, a) where a local minimum of CUI (down to -3000 m$^3$·s$^{-1}$·km$^{-1}$) was detected. By contrast, weak north-easterly winds induced a general surface flow to the west during UPW-1 (Figure 4, d). Two local cores of UPW (ranging from 1000 to 1500 m$^3$·s$^{-1}$·km$^{-1}$) could be observed: one in the south-western French coast and a secondary one nearby Cape Matxitxako. The rest of DOW events denoted in Figure 4-a (not shown) were very similar to the DOW-1 case here exposed (Figure 4, c), sharing common circulation features and comparable spatial distributions of CUI. Most of the surface current patterns were clearly related to specific wind patterns that are recurrent in the study area, in line with Borja et al. (2008) and Solabarrieta et al. (2015).

## 5 Conclusions

Over the last decades, relevant efforts have been dedicated to the indirect quantification of the intensity, duration and variability of UPW since it seriously impacts on biogeochemical cycles and ecosystem productivity (Lachkar and Gruber, 2011). Diverse CUIs have been traditionally derived from wind or sea level pressure estimations but they present some limitations such as the omission of the direct influence of ocean circulation (Jacox et al., 2018). In the present contribution, the attention was placed on the development of a purely ocean-based CUI for NWI and BOB regions (Figure 1, a), constructed from reliable hourly surface current maps provided by two different HFR remote-sensing systems.

In this context, the proposed CUI-HFR presents additional advantages with respect to previous traditional CUIs, namely:

i) it takes into consideration the direct influence of coastal waters dynamics, providing thereby a more complete portrait of this phenomenon.

ii) it provides high-resolution two-dimensional maps that can aid to elucidate the spatial distribution and magnitude of the coastal UPW together with the potential existence of recurrent patterns and/or filaments in intricate regions with complex-geometry configurations. The small-scale belt of UPW, confined in shallower coastal areas and evidenced in Figure 3 (a, c), is consistent with HFR-derived maps of horizontal divergence previously published in Lorente et al. (2020). In this previous work, it was suggested that positive divergence, localized at the tip of Cape Finisterre, induced topographic UPW and then upwelled waters were advected southwards away from the promontory. Similar initiatives with HFR current observations were effectively addressed in the west coast of the USA (Roughan et al., 2005), proposing that confined areas of semi-persistent UPW were not due to local or remote wind forcing but rather to the divergence of the prevailing southerly flow as it passed the Point Loma headland.

iii) it is generated from consistent remote-sensed hourly surface current observations, not from coarse-resolution atmospheric forecast which are probably affected by higher uncertainties. This interpretation is supported by the fact that operational atmospheric and ocean models include assimilation schemes where remote observations are routinely ingested to improve their predictive skills (Wilczak et al., 2019; Hernández-Lasheras et al., 2021).

To assess the credibility of the proposed CUI-HFR indexes, they were spatially averaged over specific subdomains (Annex 1) and later validated against CUI-WIND time series, based on hourly wind observations collected by two in situ buoys moored within each HFR footprint, during two distinct time periods: in NWI for August-December 2021 (Figure 2) and in BOB for April-August 2014 (Figure 4), respectively. The results obtained evidenced the significant agreement between CUI-HFR and CUI-WIND, with correlation coefficients of 0.80 (Annex 2, c) and 0.68 (Annex 2, e), respectively, for these two pilot areas. Furthermore, CUI-HFRs were able to adequately identify a variety of UPW and DOW episodes, regardless of their length and intensity, corroborating the recurrent relationship between the surface circulation and predominant wind patterns in NWI (Herrera et al., 2005) and BOB (Solabarrieta et al., 2015). Thanks to the computation of Hovmöller diagrams at selected transects, coastal promontories such as Cape Finisterre (NWI) and Cape Matxitxako (BOB) were revealed as important modulators of UPW processes, in line with previous works that acknowledged that capes and bays may induce significant variations in wind stress (Torres and Barton, 2006; Enriquez and Friehe, 1995).

The strong link between HFR-derived wind-induced circulation and the modulation of surface fields of temperature and chlorophyll (provided by two independent satellite missions) was also shown in the NWI area, coherent with prior literature (Picado et al., 2013). Intense northerly winds and the general flow induced to the southwest are closely associated with the cooling and fertilization of coastal waters with nutrients, fuelling high primary production that ultimately supports the notable biological diversity in this region. Within this framework, future investigations should explore the connection between ocean dynamics and marine communities and how the latter respond to persistent changes in the former. This could be achieved by analysing the relationship between the proposed CUI-HFRs and food availability, which is often represented by recruitment indexes (Borja et al., 2006).

As a proof-of-concept investigation, the prognostic capabilities of the GLOBAL analysis and forecasting system to accurately reproduce UPW/DOW events in the NWI area were also assessed (Figures 2 and 3). Following the same

approach, CUI-GLOBAL demonstrated a significant consistency, with annual correlation coefficients of 0.74 and 0.72 for 2020 (Annex 3) and 2021 (Figure 2, a). From a qualitative perspective, the daily maps of surface circulation and spatial patterns of CUI-GLOBAL presented similarities with those derived from HFR remote-sensed observations, namely: i) the rather uniform S-SW surface circulation and the related offshore deflection of the main flow during UPW-favourable conditions (Figure 3, a-d); ii) the narrow surface poleward flow along the NWI shelf edge during DOW-favourable conditions (Figure 3, e-f). The discrepancies detected between both CUI-HFR and CUI-GLOBAL maps in this specific subregion could be attributed to the fact that coastal and shelf phenomena are still poorly replicated or even misrepresented as the model grid mesh is too coarse (e.g. nominal 1/12°). This is especially true for complex-geometry regions like semi-enclosed coastal embayments where the coastline, seamounts, and bottom topography are not well resolved. In this context, mixing schemes, river inflows, and atmospheric forcings have been traditionally identified as areas of further research in global ocean modelling (Holt et al., 2017). Notwithstanding, this comparison potentially opens future pathways for the design of fit-for-purpose metrics that quantify the model skill in reproducing specific ocean processes and their seasonal variability.

Therefore, the obtained outcomes support the potential implementation of a predictive application to infer UPW/DOW conditions even ten days ahead, which is the maximum forecast horizon provided by the GLOBAL ocean model. Complementarily, the methodology proposed in this work could be also applied to the GLOBAL ocean physics reanalysis product (which covers the period 1 January 1993 – 31 December 2020) to analyse long-term seasonal trends in both study areas as extremely active and persistent UPW and DOW episodes might impact negatively on coastal ecosystems. During periods of increased offshore advection, some fish and invertebrate populations are exported from coastal habitats and exhibit reduced recruitment success (Bjorkstedt and Roughgarden, 1997). Moreover, an excessive enrichment of surface waters inshore may support the proliferation of harmful algal blooms (Pitcher et al., 2010). By contrast, the opposite-phase circulation patterns during DOW-favourable wind conditions might be related to the transport and retention of pollutants onto the shoreline, with subsequent biological and socioeconomic consequences.

In summary, results seem to suggest that the HFR performances were sound and credible for the two distinct periods and areas analysed, providing reliable surface current estimations that could be effectively used for coastal monitoring and the characterization of recurring UPW and DOW episodes. Future research endeavours should include the synergetic combination of HFRs and GLOBAL model products to implement a valuable long-term Ocean Monitoring Indicator that might be used over any coastal area of the global ocean for wise decision making to mitigate adverse effects of climate change.

**Data availability**

The model and observation products used in this study from both the Copernicus Marine Service and other sources are listed in Table 1.

**Author contributions**

PL, AR, ER, LS, JT, SP and JM conducted the pilot study through fruitful discussions in the framework of working team meetings. PL, SP, LS and ER created the figures. PL prepared a first version of the draft with inputs from all co-authors. AR

y LS analysed HF radar and in situ observations from Bay of Biscay, while ER, PL and SP provided expertise in the definition of upwelling indices. All authors participated in the iterations and revision of the manuscript.

**Competing interests**

The contact author has declared that none of the authors has any competing interests. Disclaimer. Publisher's note: Copernicus Publications remains neutral with regard to jurisdictional claims in published maps and institutional affiliations.

**Acknowledgments**

This study has been developed in the framework of the Work Package WP3 of EuroSea project (https://eurosea.eu/), which is aimed at improving and integrating the European ocean observing and forecasting system. It is also worth mentioning that this work builds upon a previous proof-of-concept investigation conducted within the context of RADAR_ON_RAIA project (Interreg V, a Spain–Portugal program). Finally, the authors are grateful to INTECMAR—Consellería do Mar—Xunta de

Galicia and the Hydrographic Institute of Portugal for the strong cooperation to jointly operate the Galician HFR network.

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

**Tables**

| Product ref. no. | Product ID & type | Data access | Documentation |
|---|---|---|---|
| 1 | INSITU_GLO_PHY_UV_DISCRETE_NRT _013_048, in situ observations | EU Copernicus Marine Service Product (2022a) | PUM: Verbrugge et al. (2022a); QUID: Verbrugge et al. (2022b) |
| 2 | INSITU_GLO_PHY_UV_DISCRETE_MY _013_044, in situ observations | EU Copernicus Marine Service Product (2022b) | PUM: Etienne et al. (2022); QUID: Etienne et al. (2023) |
| 3 | INSITU_IBI_PHYBGCWAV_DISCRETE_ MYNRT_013_033, in situ observations | EU Copernicus Marine Service Product (2022c) | PUM: In situ TAC partners (2022); QUID: Wehde et al. (2022) |
| 4 | GLOBAL_ANALYSISFORECAST_PHY_0 01_024, numerical models | EU Copernicus Marine Service Product (2022d) | PUM: Le Galloudec (2022); QUID: Lellouche et al. (2022) |
| 5 | SST_GLO_SST_L4_REP_OBSERVATION S_010_011, satellite observations | EU Copernicus Marine Service Product (2022e) | PUM: Worsfold et al. (2022); QUID: Worsfold et al. (2023) |
| 6 | OCEANCOLOUR_GLO_BGC_L4_MY_00 9_104, satellite observations | EU Copernicus Marine Service Product (2022f) | PUM: Colella et al. (2022); QUID: Garnesson et al. (2022) |

**Table 1. Products from the Copernicus Marine Service used in this study, including the Product User Manual (PUM) and QUality Information Document (QUID). Last access for all web pages cited in this table: 2 June 2023.**

| Name (area) | Type | Deployment | Longitude | Latitude | Depth | Sampling | Coriolis parameter |
|---|---|---|---|---|---|---|---|
| Silleiro (NWI) | Seawatch | 1998 | 9.44ºW | 42.12ºN | 600 m | 1 h | $9.75 \cdot 10^{-5}$ s$^{-1}$ |
| Bilbao (BOB) | Seawatch | 1990 | 3.04ºW | 43.64ºN | 580 m | 1 h | $10 \cdot 10^{-5}$ s$^{-1}$ |

**Table 2. Description of the buoys (product ref. no. 3 in Table 1) deployed within the coverage of each HFR system (product ref. no. 1 and no. 2 in Table 1), as shown in Figure 1. More information on products can be found in the text and in the product table.**

**Figures**

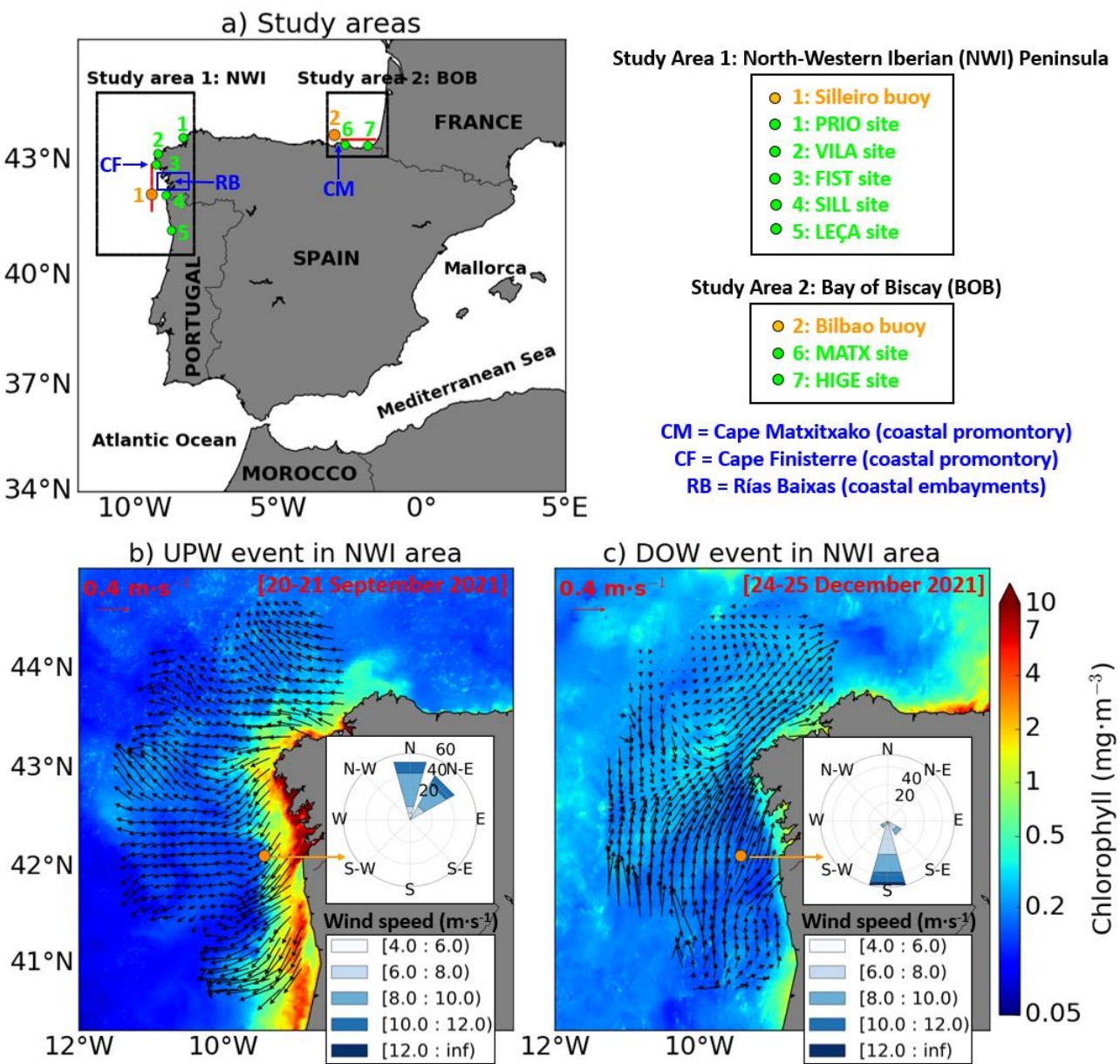

Figure 1. a) Study areas are marked with black squares: the North-Western Iberian (NWI) Peninsula upwelling system and the Bay of Biscay (BOB), where two High-Frequency radars (HFRs) are deployed -product ref. no. 1 and no. 2 (Table 1)-. Green dots denote HFR sites, while orange dots represent Silleiro buoy and Bilbao buoy locations -product ref. no. 3 (Table 1)-, moored within NWI and BOB areas, respectively. Red lines indicate selected transects; HFR-derived surface circulation and satellite-derived chlorophyll distribution -product ref. no. 6 (Table 1)- during a 2-day (b) upwelling summer event and (c) downwelling winter event in the NWI area, when intense northerly and southerly winds were predominant, respectively.

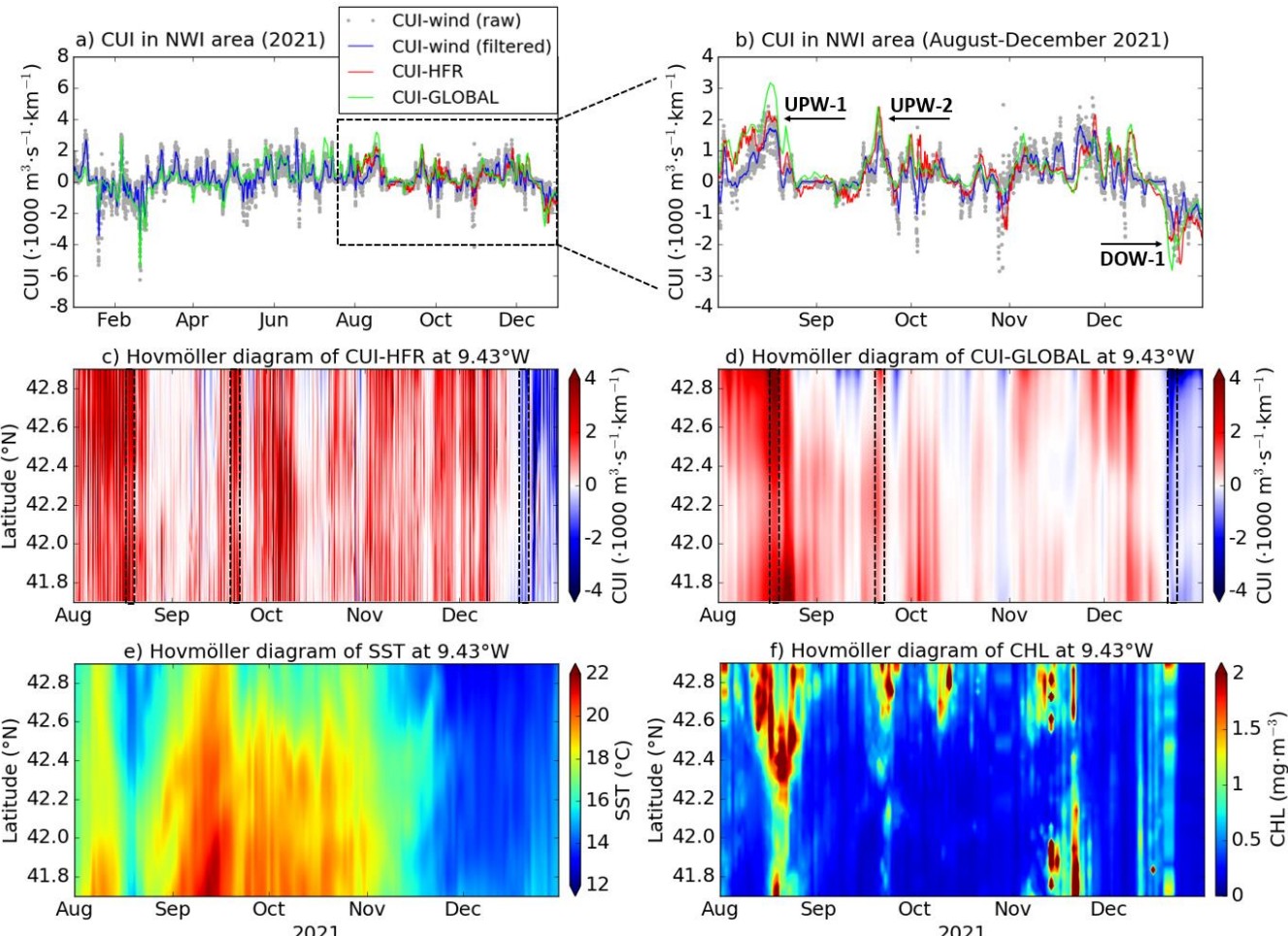

**Figure 2. Time Series of diverse hourly Coastal Upwelling Index (CUI) in the North-Western Iberia (NWI) region (a) for the entire 2021 and (b) for August-December 2021, as derived from wind observations from Silleiro Buoy (CUI-WIND) -product ref. no. 3 (Table 1)- , from HFR surface currents (CUI-HFR) - product ref. no. 1 and no. 2 (Table 1) - and from modelled surface currents (CUI-GLOBAL) -product ref. no. 4 (Table 1)-, respectively. CUI-WIND raw (grey dots) was filtered by applying a 24 h moving mean (blue line). Hovmöller diagrams of (c) CUI-HFR, (d) CUI-GLOBAL, (e) sea surface temperature (SST) -product ref. no. 5 (Table 1)- and (f) chlorophyll (CHL) -product ref. no. 6 (Table 1)- concentration during the period August-December 2021 for a selected transect of constant longitude (9.43ºW), depicted in Figure 1-a.**

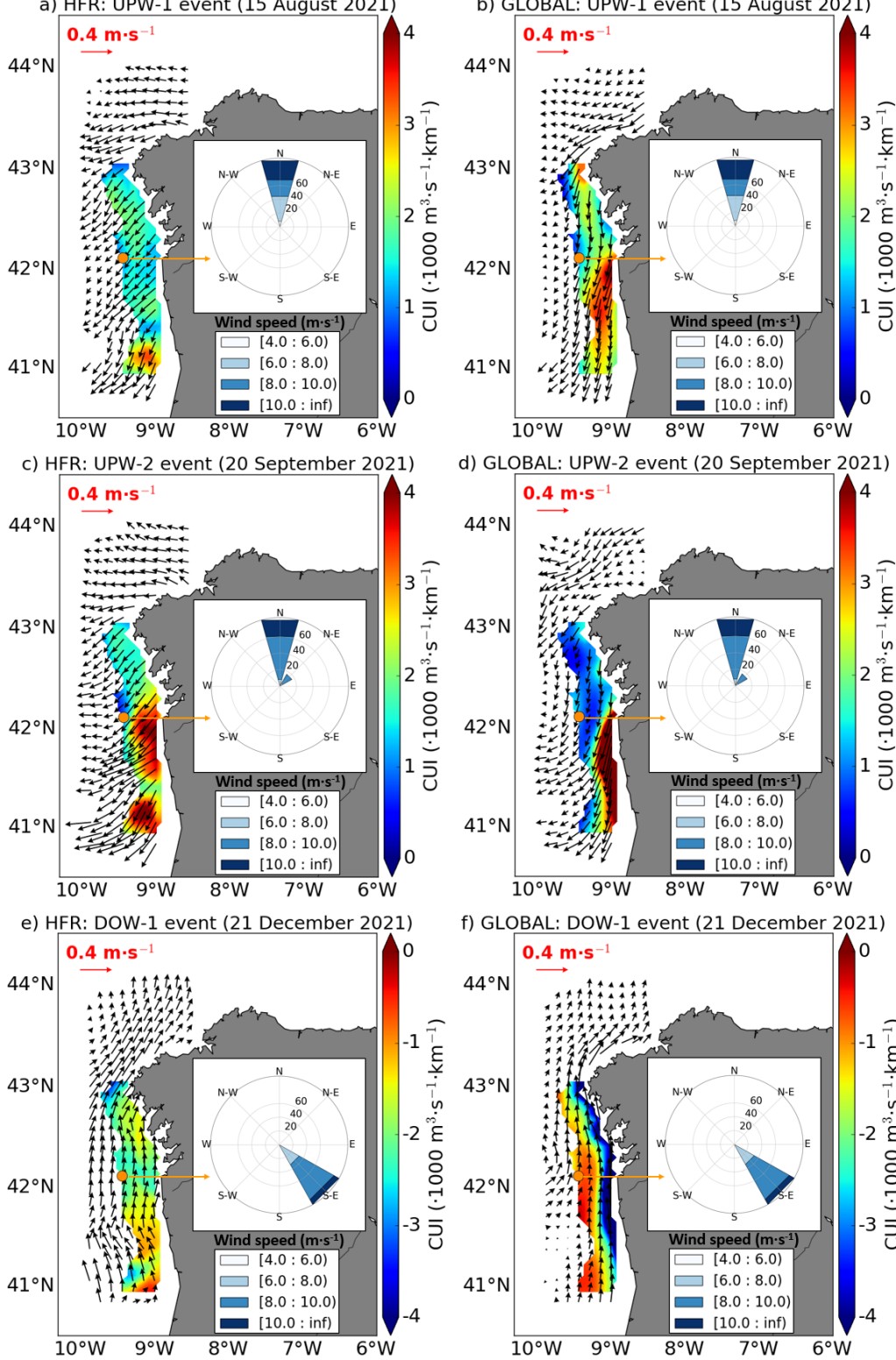

**Figure 3. Daily-averaged maps of circulation and Coastal Upwelling Index (CUI) as derived from surface current estimations provided by a High Frequency radar (HFR, left column, -product ref. no. 1 and no. 2 in Table 1-) and GLOBAL model (right column, -product ref. 4-) for each specific upwelling (UPW) and downwelling (DOW) episode categorized in Figure 2-b. Daily wind roses derived from hourly wind estimations from Silleiro Buoy (represented by an orange dot, -product ref. no. 3 in Table 1-) are also provided, indicating incoming wind direction and speed.**

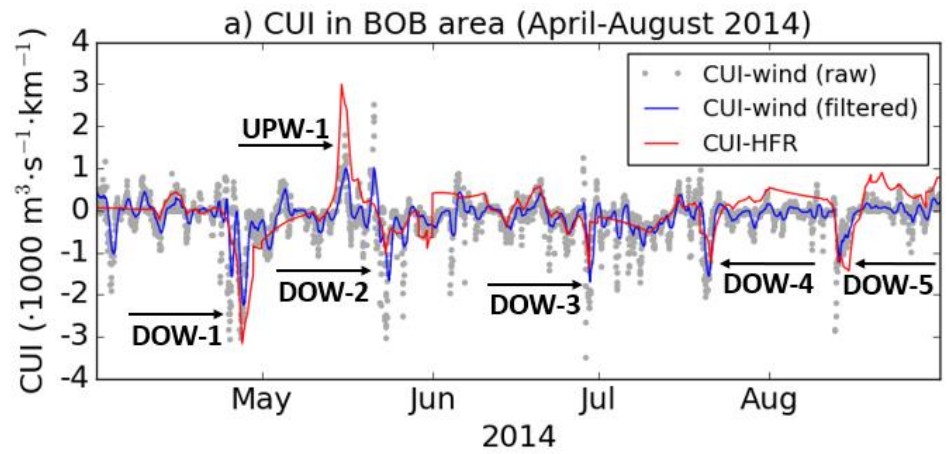

a) CUI in BOB area (April-August 2014)

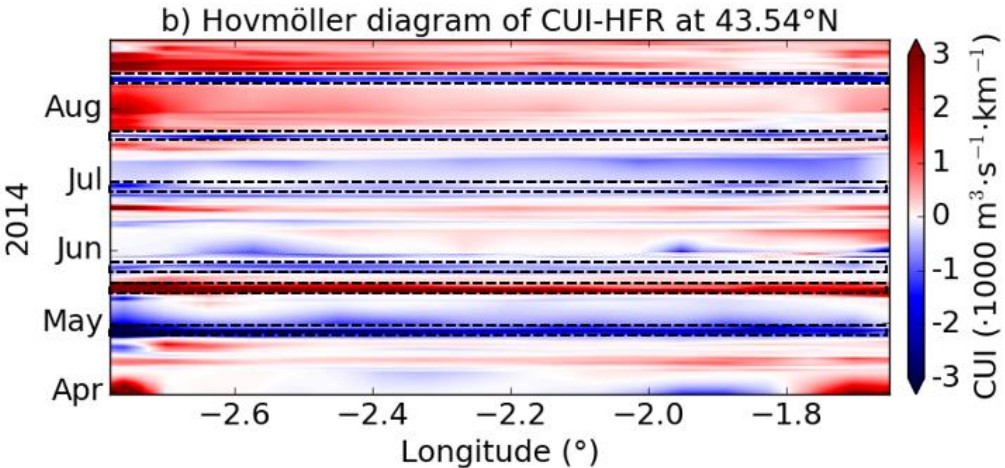

b) Hovmöller diagram of CUI-HFR at 43.54°N

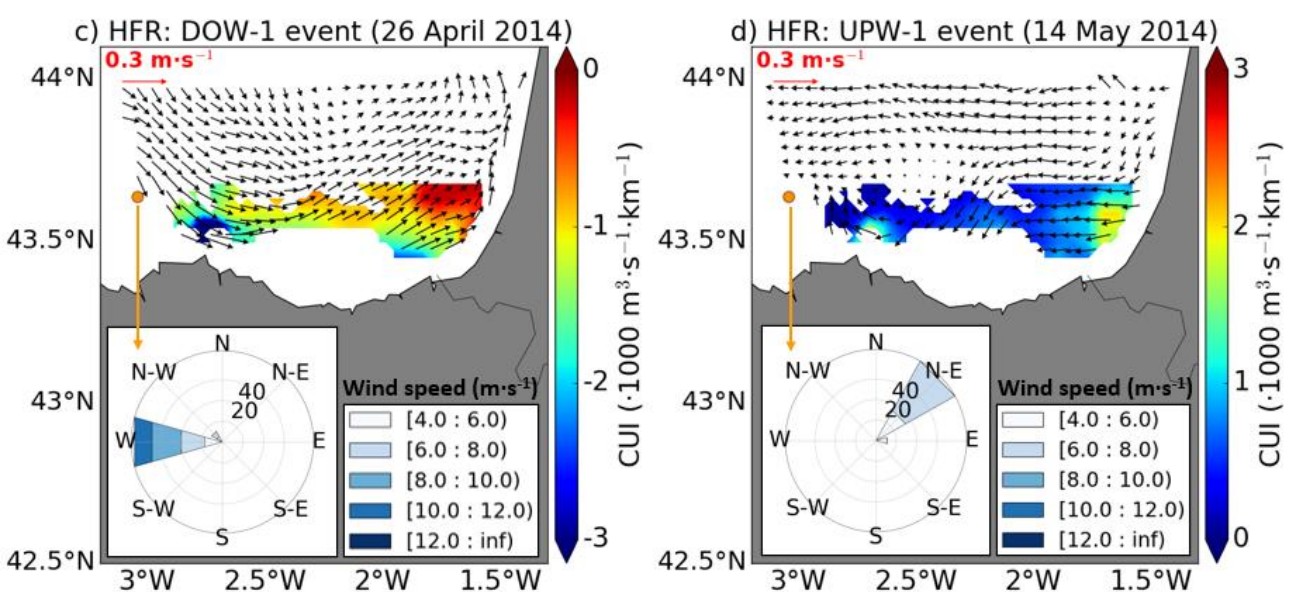

c) HFR: DOW-1 event (26 April 2014)

d) HFR: UPW-1 event (14 May 2014)

**Figure 4. a) Coastal upwelling indexes (CUI) in the Bay of Biscay (April-August 2014), based on hourly wind observations (from Bilbao buoy, -product ref. no. 3 in Table 1-) and hourly surface currents (from an HFR system, -product ref. no. 1 and no. 2 in Table 1-). CUI-WIND raw (grey dots) was filtered by applying a 24 h moving mean (blue line). Identification of 6 UPW/DOW events; b) Hovmöller diagram of CUI-HFR for a transect of constant latitude, where UPW (DOW) events are marked with black dotted squares; c-d) Daily averaged maps of surface circulation and CUI-HFR for 2 events. Daily wind roses from Bilbao buoy (represented by an orange dot, product ref. no. 3 in Table 1) are also provided, indicating incoming wind direction and speed.**

**Annex**

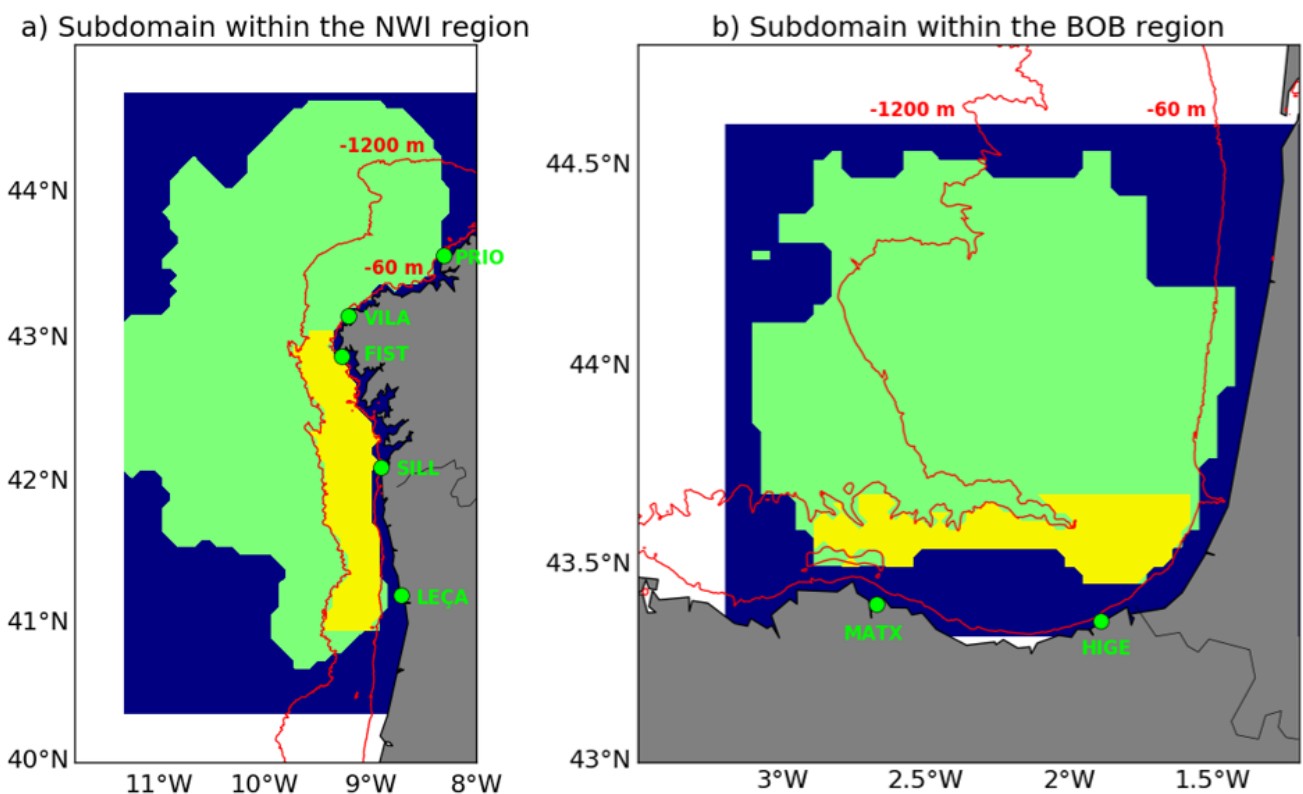

**Annex 1. Selection of subdomains within each HFR coverage -product ref. no. 1 and no. 2 (Table 1)-. Blue area represents the entire HFR spatial grid, while the green area denotes the subdomain where the value of the dimensionless GDOP parameter is below 3. Yellow area is the selected HFR subdomain to compute the coastal upwelling index (CUI) and must fulfil two requirements: i) GDOP must be below 3; and ii) it must be comprised between bathymetric contours (red lines) of 60 and 1200 m depth. Green dots represent the HFR sites that compose each network deployed in the NWI (a) and BOB (b) regions.**

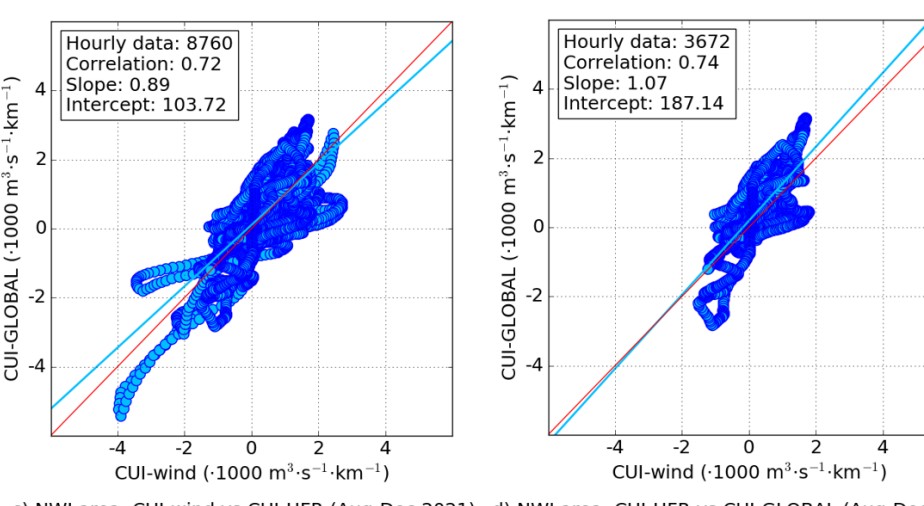

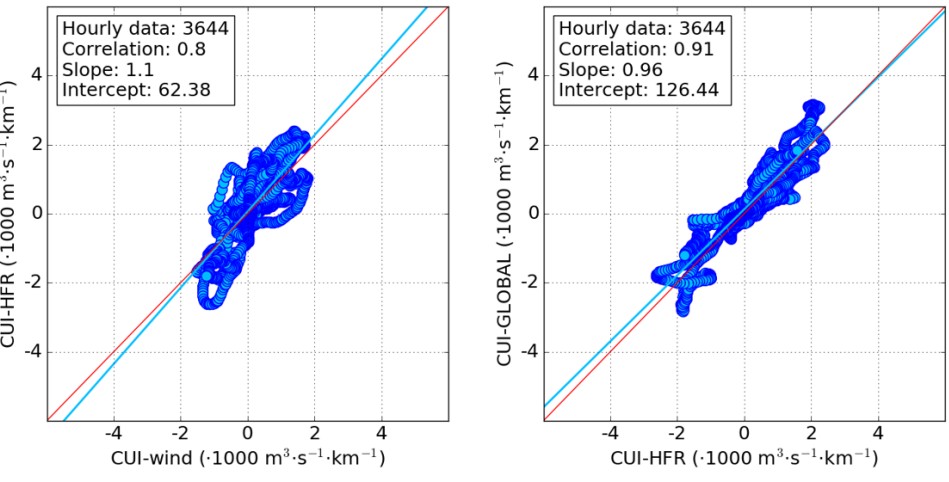

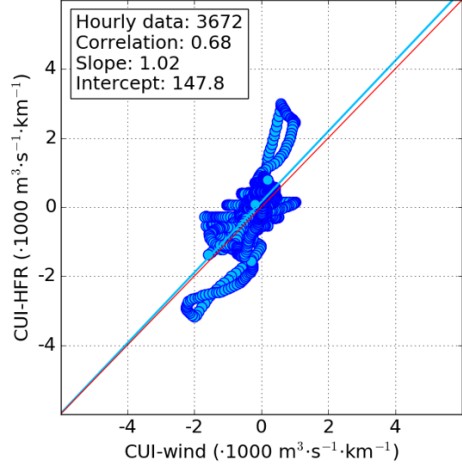

**Annex 2. Best linear fit of scatter plots between: (a-b) CUI-GLOBAL (product ref. no. 4 in Table 1)- and CUI-WIND (filtered) -product ref. no. 3 (Table 1)- in NWI area for the entire 2021 and August-December 2021; (c) CUI-HFR - product ref. no. 1 and no. 2 (Table 1)- and CUI-WIND (filtered) -product ref. no. 2 (Table 1)- in NWI area for August-December 2021; (d) CUI-GLOBAL -product ref. no. 4 (Table 1)- and CUI-HFR -product ref. no. 1 and no. 2 (Table 1) in NWI area for August-December 2021; (e) CUI-HFR -product ref. no. 1 and no. 2 (Table 1)- and CUI-WIND (filtered) -product ref. no. 3 (Table 1)- in BOB area for April-August 2014; Statistical metrics gathered in white boxes.**

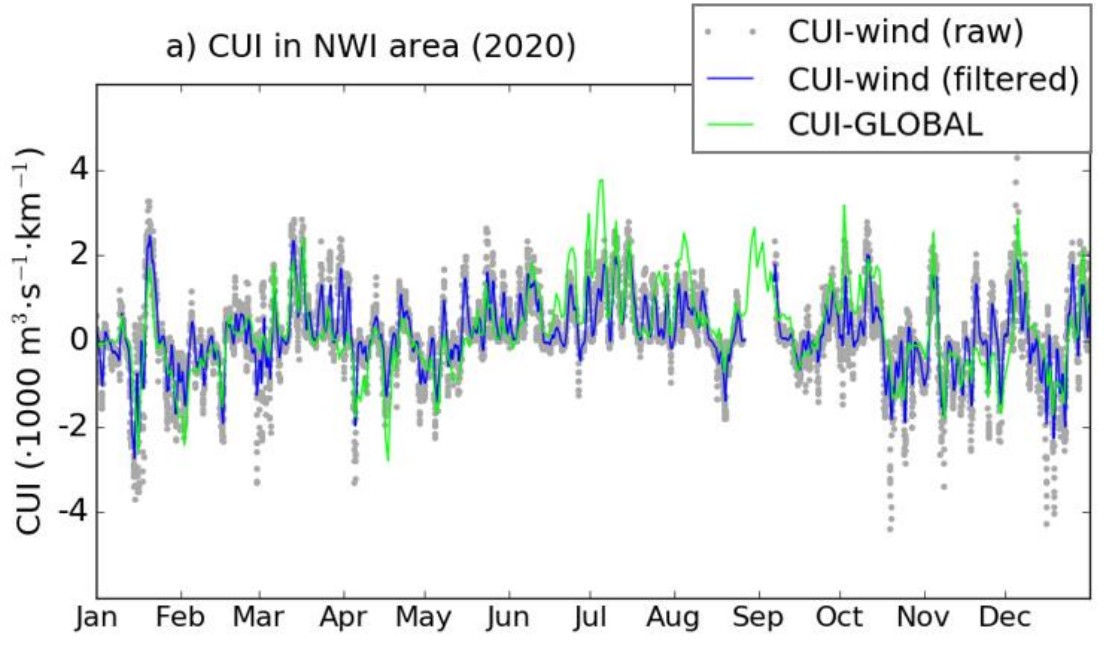

a) CUI in NWI area (2020)

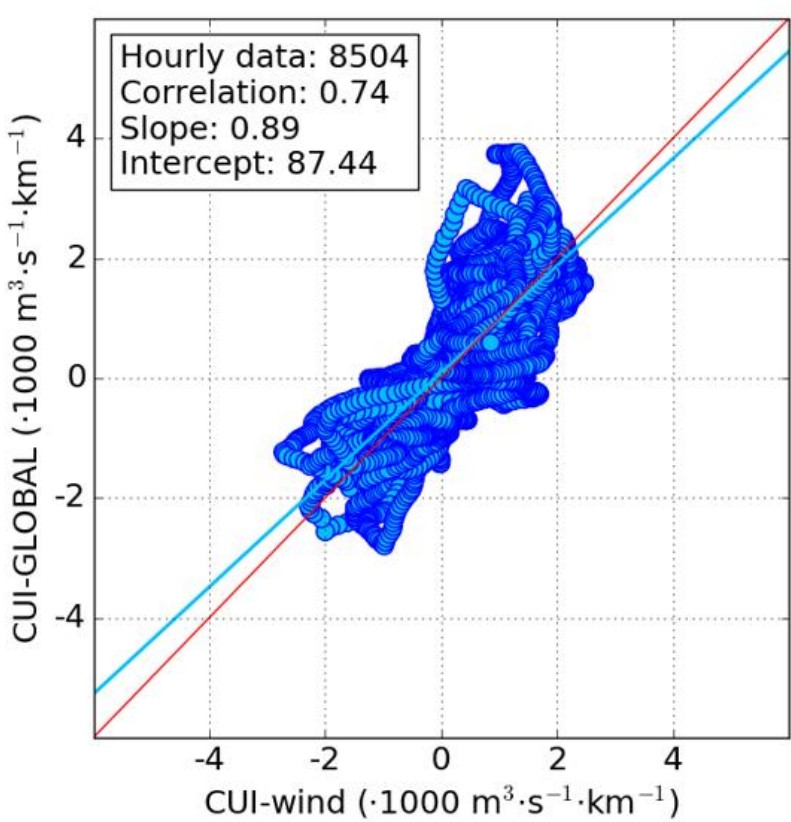

b) NWI area: CUI-wind vs CUI-GLOBAL (2020)

Hourly data: 8504
Correlation: 0.74
Slope: 0.89
Intercept: 87.44

**Annex 3. (a) Time Series of hourly Coastal Upwelling Index (CUI) in the North-Western Iberia (NWI) region for the entire 2020 as derived from wind observations from Silleiro Buoy (CUI-WIND) -product ref. no. 3 (Table 1)- and from modelled surface currents (CUI-GLOBAL) -product ref. no. 4 (Table 1)-. CUI-WIND raw (grey dots) was filtered by applying a 25 h moving mean (blue line); (b) Best linear fit of scatter plot between CUI-GLOBAL and**
720 **CUI-WIND (filtered). Statistical metrics gathered in the white box.**
