# Peer review of "High Frequency radar-derived coastal upwelling index"

_State of the Planet, 2022_

## Referee Comment (RC1)

This paper proposes using surface current maps derived from High Frequency Radars (HFR) to define a Coastal Upwelling Index (CUI), a quantity that is usually obtained from other met-ocean parameters such as wind speed, sea level pressure field or Sea Surface Temperature (SST). The advantage of the HFR-derived CUI (CUI-HFR) over classical indices is the ability to provide spatial maps (instead of mere time series) and to account for the ocean circulation which cannot be apprehended by the other parameters. To assess the relevance of this new index, the data originating from 2 networks of HFR in the North-West Iberian Peninsula are processed and compared with the traditional wind-derived CUI (CUI-WIND) as well as with a CUI defined from a global operational 3D ocean model (CUI-GLOBAL). In addition, some satellite measurements of SST and Clorophyll are used to corroborate the upwelling/downwelling events.

The paper is interesting, well written and well documented. It introduces a promising and important new application of HFR.  For these reasons I think it deserves acceptance for publication. I have only a list of minor remarks and questions, whose clarification could help consolidating the methodology and results. I list them below in  order of appearance in the text.

1) Section 4 p 7 : ocean-based CUI and wind-based CUI are found strongly correlated. Does this merely mirrors the correlation between winds and surface currents or is there a deeper reason ? What would be the correlation coefficient between the components of wind and current velocities (u-wind versus u-current and v-wind versus v-current) ?

2) I see no statistical comparison (correlation coefficient, RMS difference) to compare CUI-GLOBAL and CUI-HFR. This would be interesting to see how close they in order to coarsely quantify  the accuracy that can be expected from these two types of estimators.

3) What is the influence of tidal currents on the hourly CUI-HFR ? As the CUI-GLOBAL is free of tide, I suppose this induces an extra difference ?

4) p 8 line 228 : The « overall concordance » between HFR-CUI and GLOBAL-CUI seems somewhat euphemistic when looking at Figure 3. There are some important differences both in magnitude and direction of the currents. Is there any clue as to which data (HFR or model) is more reliable ?

5) For the upwelling event #2 (Figure3 c), the HFR data around latitude 41.4 N show a localized drop of intensity of the CUI-HFR which is not consistent with the model (Figure 3d) The same phenomenon is visible for the upwelling event #1 although less pronounced (Figure 3a). At first sight this could be interpreted as a systematic error of the HFR measurement in this aera. On the other hand, the Chlorophyll map on Figure 1a)  shows the same disconnected structures around the same latitude, supporting the HFR pattern. Could you comment on this qualitative difference between HFR and GLOBAL in this case ?  What can be said on the reliability of HFR measurement around this small area ?

6) p 9 line 280 : « with » respect to

---

## Author Response (AR1)

**REVIEWER-1**

This paper proposes using surface current maps derived from High Frequency Radars (HFR) to define a Coastal Upwelling Index (CUI), a quantity that is usually obtained from other met-ocean parameters such as wind speed, sea level pressure field or Sea Surface Temperature (SST). The advantage of the HFR-derived CUI (CUI-HFR) over classical indices is the ability to provide spatial maps (instead of mere time series) and to account for the ocean circulation which cannot be apprehended by the other parameters. To assess the relevance of this new index, the data originating from 2 networks of HFR in the North-West Iberian Peninsula are processed and compared with the traditional wind-derived CUI (CUI-WIND) as well as with a CUI defined from a global operational 3D ocean model (CUI-GLOBAL). In addition, some satellite measurements of SST and Chlorophyll are used to corroborate the upwelling/downwelling events.

The paper is interesting, well written and well documented. It introduces a promising and important new application of HFR. For these reasons I think it deserves acceptance for publication. I have only a list of minor remarks and questions, whose clarification could help consolidate the methodology and results. I list them below in order of appearance in the text.

Many thanks to Dr. Guèrin for the detailed review and the number of useful tips provided. Please find below a thorough point-by-point response with the hope of improving the quality of the document to make it acceptable for final publication.

1) Section 4, page 7: ocean-based CUI and wind-based CUI are found strongly correlated. Does this merely mirror the correlation between winds and surface currents or is there a deeper reason?

Yes, indeed the wind-surface current linear correlation is one of the premises for computing the constant parameter of the CUI (as exposed in section 3, page 6), which assume that the alongshore wind stress is the primary driver of upwelling circulation and that HF radar-derived surface currents are highly responsive to local wind (e.g. Paduan and Rosenfeld, 1996).

Nevermind, it is worth clarifying that in equation [1] of this manuscript (page 6) the wind velocity components (u and v) are measured at one single point (Silleiro buoy location, 9.43ºW, 42.12ºN) and used here as a proxy for the local open sea wind conditions. By contrast, in equation [3], the HFR-derived surface current velocities (u and v) are measured over a selected subregion (the yellow area in Annex 1-a) and then spatially averaged.

To better clarify this point, we have added the following paragraph to the Methodology section: "**Assuming the prompt and direct reaction of the upper ocean layer to intense and prolonged wind forcing in NWI (Herrera et al., 2005), it seems reasonable to develop an ocean-based indicator for UPW and DOW conditions**. Analogously, the CUI-HFR is defined as follows:"

References:

Paduan, J. D., and Rosenfeld, L. K. (1996), Remotely sensed surface currents in Monterey Bay from shore-based HF radar (Coastal Ocean Dynamics Application Radar), J. Geophys. Res., 101(C9), 20669– 20686, doi:10.1029/96JC01663.

Herrera, J.L.; Piedracoba, S.; Varela, R.; Rosón, G. Spatial analysis of the wind field on the western coast of Galicia (NW Spain) from in situ measurements. Cont. Shelf Res. 2005, 25, 1728–1748.

What would be the correlation coefficient between the components of wind and current velocities (u-wind versus u-current and v-wind versus v-current)?

Please find below the hourly timeseries (for the concurrent period 1 August - 31 December 2021) and the correlation coefficient derived from the best linear fit of scatter plot, not only for the zonal (u) and meridional (v) velocities but also for the total velocity. As explained later in this document, a 25-h running-mean filter was applied to the wind and current velocities.

As it can be observed below, the zonal component is just moderately correlated (0.43), whereas the meridional component appears to be highly correlated (0.80) for the 5-month period. This confirms our previous statement: the alongshore wind (v) stress is the primary driver of upwelling circulation in the NWI area. Finally, the total wind speed observed at Silleiro buoy and the HFR-derived total current velocity (averaged over the aforementioned subregion) are also significantly correlated, with a correlation coefficient of 0.71.

Although we have not added the figure below to the manuscript (due to the journal tight space limitations), we have commented the results derived from the scatterplot between V-wind and V-currents, with a high correlation coefficient of 0.80.

To better clarify this point, we have added the following paragraph to the beginning of Results section: "According to the statistical results exposed in Annex 2 (a-d), we can state: i) the slope and intercept values were close to 1 and moderately low, respectively; **and ii) ocean-based CUI and CUI-WIND are strongly correlated, likely due to the role of alongshore wind stress as primary driver of UPW conditions in the NWI area. Hourly alongshore winds (from Silleiro buoy) and HFR-derived alongshore currents are highly correlated (0.80) for August-December 2021 (not shown)**".

[Figure]

a) Zonal velocity (August-December 2021)

U$_{current}$ **multiplied by 97.2**
**to get a slope=1.0 and:**
**Correlation = 0.43**
**Intercept = -3.61**

b) Meridional velocity (August-December 2021)

V$_{current}$ **multiplied by 59.2**
**to get a slope=1.0 and:**
**Correlation = 0.80**
**Intercept = -0.55**

c) Total velocity (August-December 2021)

**speed**$_{current}$ **multiplied by 72.5**
**to get a slope=1.0 and:**
**Correlation = 0.71**
**Intercept = 4.26**

\* HFR-derived currents are expressed in cm/s

2) I see no statistical comparison (correlation coefficient, RMS difference) to compare CUI-GLOBAL and CUI-HFR. This would be interesting to see how close they are in order to coarsely quantify the accuracy that can be expected from these two types of estimators.

We fully agree that the statistical comparison between CUI-GLOBAL and CUI-HFR could be beneficial, so we have complemented the Annex-2 figure by adding a new panel d). In this new panel (shown below), the best linear fit of the scatterplot for a 5-month period (August-December 2021) and the associated metrics show the significant agreement between both estimators, as reflected by a correlation coefficient of 0.91 and a slope (0.96) rather close to 1.

[Figure]

d) NWI area: CUI-HFR vs CUI-GLOBAL (Aug-Dec 2021)

Hourly data: 3644
Correlation: 0.91
Slope: 0.96
Intercept: 126.44

Therefore, we have modified the following paragraph in Section 4 Results: "The visual resemblance between the three different CUIs was noticeable, with significantly high correlation coefficients: i) 0.72 and 0.74 between CUI-GLOBAL and CUI-WIND for the entire 2021 and for August-December 2021, respectively, as reflected by their best linear fit of scatter plots (Annex 2, a-b); ii) 0.80 between CUI-HFR and CUI-WIND for August-December 2021 (Annex 2, c); **iii) 0.91 between CUI-GLOBAL and CUI-HFR for August-December 2021 (Annex 2, d)."**

Additionally, we have moved panel e) of Annex-2 figure to a new figure (Annex-3, focused on 2020 year), which has also been complemented with the related scatterplot and metrics, highlighting again the potential of CUI-GLOBAL as a proxy of upwelling conditions in the study area.

[Figure]

Annex 3: a) Time Series of two hourly Coastal Upwelling Index (CUI) in the North-Western Iberia (NWI) region (a) for the entire 2020 as derived from wind observations from Silleiro Buoy (CUI-WIND) and from modelled surface currents (CUI-GLOBAL). CUI-WIND raw (grey dots) was filtered by applying a 25-h moving mean (blue line). b) Best linear fit of scatter plots between CUI-GLOBAL and CUI-WIND (filtered) in NWI area for the entire 2020.

3) What is the influence of tidal currents on the hourly CUI-HFR? As the CUI-GLOBAL is free of tide, I suppose this induces an extra difference?

Running-mean filters are the simplest low-pass filters to apply to ocean observations with the aim of: i) visually smoothing the time-series graph; and ii) revealing an outline of longer-period variations (Shirakata et al., 2016). Within this context and considering the semi-diurnal nature of the tide in the area of study, most of the tidal signal was filtered out as HFR current hourly data fields were averaged over 25-h hours to remove the main diurnal and semidiurnal tidal constituents, particularly the M2 signal, which is the largest harmonic constituent in the study area. Shorter digital filters (like the one used in this work) are generally preferred because they can yield a longer output time series from a practical finite input time series with less loss of length at both ends of the input series.

Although the 25-h running-mean filter leaves fortnightly, monthly, semi-annual tidal components (e.g. $M_f$, $M_m$, $S_{sa}$, etc.), they are considered to play a marginal role in the differences detected between CUI-HFR and CUI-GLOBAL. On the contrary, the discrepancies between both CUIs are supposed to be attributable to a major extent to the intrinsic limitations of the global ocean model and the uncertainties in the HFR derived current measurements.

To better clarify this point, a sentence has been added to the manuscript, in particular, in section 3 ("Methodology"): "[…] where u and v represent the **filtered** hourly time series of zonal and meridional surface current velocities (m·s−1) provided by the HFR, respectively. **A 25-h running-mean filter was used to smooth time-series data by suppressing the main diurnal and semidiurnal tidal constituents (Shirakata et al., 2016), particularly the M2 signal, which is the largest harmonic constituent in the study area**."

Reference:

Shirahata, K., Yoshimoto, S., Tsuchihara, T. and Ishida, S. Digital Filters to Eliminate or Separate Tidal Components in Groundwater Observation Time-Series Data. Japan Agricultural Research Quarterly (JARQ), 2016, Vol. 50, Issue 3, 241-252, doi:10.6090/jarq.50.241.

4) page 8 line 228: The «overall concordance» between HFR-CUI and GLOBAL-CUI seems somewhat euphemistic when looking at Figure 3. There are some important differences both in magnitude and direction of the currents. Is there any clue as to which data (HFR or model) is more reliable?

We agree with the reviewer that the surface circulation maps present some differences that must be further discussed along with their source (e.g. limitations of the global model at coastal scales, littoral processes misrepresented, etc.).

However, our main intention was to indicate that, despite those discrepancies in magnitude and direction, both the HFR-derived and GLOBAL-derived maps in Figure 3 share some common features at synoptic scale, namely: i) the prevailing S-SW surface flow, as response to northerly winds during UPW episodes, ii) the general poleward flow (the so-called Iberian Poleward Current) along the NW Iberian shelf during DOW episodes (Figure 3, e-f). Furthermore, both types of maps present rather uniform, smooth patterns where no submesoscale structures (i.e. eddies, small meanders, vortexes, etc.) are evidenced since the wind-induced homogenization also reduced the patchiness.

To avoid any misunderstanding, we have rephrased the aforementioned sentence: "**Despite of the observed discrepancies in magnitude and direction, maps of wind-induced surface currents shared some common features: i) the prevailing S-SW surface circulation (as response to northerly winds) along with the typical offshore deflection of the flow, associated with UPW-favourable conditions; ii) the rather uniform circulation westwards (south-westwards) to the north (south) of Cape Finisterre (indicated in Figure 1, a); iii) the absence of submesoscale structures (i.e. eddies, small meanders, etc.) due to the strong wind-induced homogenization**."

With regards to the question about the reliability of HFR estimations and GLOBAL model outputs, it has been broadly accepted that quality-controlled HFR estimations should act as "ground truth". To support this assumption, a wide variety of previous works can be found in the literature where ocean forecasting models were assessed against HFR-derived surface currents, used as consistent benchmark reference to compare with (Berta et al., 2014, Lorente et al., 2021; Aguiar et al., 2020; Sotillo et al., 2021, to name a few).

Finally, we would like to provide a flavor of the basic functionalities of the NARVAL (North Atlantic Regional VALidation) software package (Lorente et al., 2019) in order to elucidate the accuracy of the GLOBAL forecast model against the Galician HFR on a monthly basis (August 2021). The figure below shows in the pop panel (from left to right): a) Monthly averaged HFR derived surface currents; b) Monthly averaged GLOBAL forecast model surface currents; c) Correlation of zonal and d) meridional surface currents between the GLOBAL forecast model and the HFR. In the bottom panel (from left to right): a) RMSE of zonal and b) meridional surface currents; c) Magnitude of the complex correlation (i.e. index) and d) phase between HFR and GLOBAL model-predicted currents.

[Figure]

As it can be seen above, the degree of concordance in coastal areas is good (particularly for the zonal component and for the complex correlation coefficients, which are both above 0.7) and moderate RMSE values (higher for the meridional component in the near-coastal areas). It is also true that the veering angle (between HFR current vectors and GLOBAL current vectors) ranges from -15º to -30º on average, indicating thereby the counter-clockwise rotation of GLOBAL vectors with respect to HFR vectors. In other words, while HFR vectors show a prevailing SW flow, GLOBAL vectors seem to represent a predominant southward flow near the coast.

References:

Berta, M., Bellomo, L., Magaldi, M. G., Griffa, A., Molcard, A., Marmain, J., Borghini, M., and Taillandier, V.: Estimating Lagrangian transport blending drifters with HF radar data and models: Results from the TOSCA experiment in the Ligurian Current (North Western Mediterranean Sea), Prog. Oceanogr., 128, 15– 29, https://doi.org/10.1016/j.pocean.2014.08.004, 2014.

Aguiar, E., Mourre, B., Juza, M., Reyes, E., Hernández-Lasheras, J., Cutolo, E., Mason, E., and Tintoré, J.: Multi-platform model assessment in the Western Mediterranean Sea: impact of downscaling on the surface circulation and mesoscale activity, Ocean Dnam., 70, 273– 288, https://doi.org/10.1007/s10236-019-01317- 8, 2020.

Lorente, P., Sotillo, M. G., Amo-Baladrón, A., Aznar, R., Levier, B., Aouf, L., Dabrowski, T., Pascual, Á. De, Reffray, G., Dalphinet, A., Toledano, C., Rainaud, R., and Álvarez-Fanjul, E.: The NARVAL Software Toolbox in Support of Ocean Models Skill Assessment at Regional and Coastal Scales, in ICCS 2019, Lecture Notes in Computer Science, edited by: Rodrigues, J., (Cham: Springer), https://doi.org/10.1007/978-3-030-22747- 0_25, 2019.

Lorente, P., Lin-Ye, J., García-León, M., Reyes, E., Fernandes, M., Sotillo, M. G., Espino, M., Ruiz, M. I., Gracia, V., Perez, S., Aznar, R., Alonso-Martirena, A., and Álvarez-Fanjul, E.: On the Performance of High Frequency Radar in the Western Mediterranean During the Record-Breaking Storm Gloria, Front. Mar. Sci., 8, 205, https://doi.org/10.3389/fmars.2021.645762, 2021.

5) For the upwelling event #2 (Figure 3c), the HFR data around latitude 41.4ºN show a localized drop of intensity of the CUI-HFR which is not consistent with the model (Figure 3d) The same phenomenon is visible for the upwelling event #1 although less pronounced (Figure 3a). At first sight this could be interpreted as a systematic error of the HFR measurement in this area. On the other hand, the Chlorophyll map on Figure 1a) shows the same disconnected structures around the same latitude, supporting the HFR pattern. Could you comment on this qualitative difference between HFR and GLOBAL in this case?

This dipole-like structure has already been observed in the CUI-HFR maps for July 2014 and reported by Lorente et al. (2020) in the same area. Furthermore, both cores were also evidenced in satellite-derived SST and CHL maps (Figure 8a in Lorente et al., 2020), in agreement with the spatial distribution of CUI-HFR.

A paragraph has been added to the manuscript to better clarify this point: "**The drop of CUI-HFR is consistent (in timing and location) with the drop of CHL concentration showed in Figure 1-b, supporting this HFR pattern which was already documented in Lorente et al. (2020)**".

It is also worth mentioning that the dipole-like structure has been observed several times at different relative positions, highlighting: i) the importance of the Galician shoreline orientation (with respect to the prevailing wind direction) in modulating upwelling features; ii) the importance of Cape Finisterre promontory. Abrupt changes in coastal orientation can induce noticeable wind stress fluctuations and, hence, different upwelling conditions with subsequent biophysical implications, as previously documented by Álvarez et al. (2005 and 2011) and Torres et al. (2003) in the same region.

The discrepancies detected between both CUI-HFR and CUI-GLOBAL maps in this specific subregion could be attributed, to a large extent, to the inherent shortcomings of a global ocean model, among others:

i) The coarse model spatial resolution (1/12º = 9.25 km), which handicaps the detection of smaller (sub)mesoscale features.

ii) The underestimation (or even the misrepresentation) of some coastal processes that could play a secondary role in the surface coastal circulation, like the freshwater discharge from Galician rivers (coincident with the upper core of cold SST, higher CHL and higher CUI-HFR). GLOBAL model includes only the impact of 100 major rivers by means of a smoothed climatology (Product User Manual, 2022)

A paragraph has been added to the manuscript to better clarify this point: "**The discrepancies detected between both CUI-HFR and CUI-GLOBAL maps in this specific subregion could be attributed to the fact that coastal and shelf phenomena are still poorly replicated or even misrepresented as the model grid mesh is too coarse (e.g. nominal 1/12º). This is especially true for complex-geometry regions like semi-enclosed coastal embayments where the coastline, seamounts, and bottom topography are not well resolved. In this context, mixing schemes, river inflows, and atmospheric forcings have been traditionally identified as areas of further research in global ocean modelling (Holt et al., 2017).**"

References:

Lorente, P., Piedracoba, S., Montero, P., Sotillo, M. G., Ruiz, M. I. and Álvarez-Fanjul, E.: Comparative Analysis of Summer Upwelling and Downwelling Events in NW Spain: A Model-Observations Approach, Remote Sens., 12(17), doi:10.3390/rs12172762, 2020.

Product User Manual (November 2022):

https://catalogue.marine.copernicus.eu/documents/PUM/CMEMS-GLO-PUM-001-024.pdf

Álvarez, I.; Gómez-Gesteira, M.; deCastro, M.; Prego, R. Variation in upwelling intensity along the North-West Iberian Peninsula (Galicia). J. Atmos. Ocean Sci. 2005, 10, 309–324.

Álvarez, I.; Gómez-Gesteira, M.; deCastro, M.; Lorenzo, M.N.; Crespo, A.J.C.; Dias, J.M. Comparative analysis of upwelling influence between the western and northern coast of the Iberian Peninsula. Cont. Shelf Res. 2011, 31, 388–399.

Torres, R.; Barton, E.D.; Miller, P.; Álvarez-Fanjul, E. Spatial patterns of wind and sea surface temperature in the Galician upwelling region. J. Geophys. Res. 2003, 108, 3130.

Holt, J., Hyder, P., Ashworth, M., Harle, J., Hewitt, H. T., Liu, H., New, A. L., Pickles, S., Porter, A., Popova, E., Allen, J. I., Siddorn, J., and Wood, R.: Prospects for improving the representation of coastal and shelf seas in global ocean models, Geosci. Model Dev., 10, 499–523, https://doi.org/10.5194/gmd-10-499-2017, 2017.

What can be said on the reliability of HFR measurement around this small area?

To ensure the consistency and reliability of HFR remote sensed estimations, an integrated approach was adopted, which consisted of: i) the real-time web monitoring of non velocity-based diagnostic parameters along with ii) regular-scheduled validation exercises of HFR data (at both the radial and total vector levels) against independent *in situ* observations in order to provide upper bounds on the radar current measurement accuracy.

Regarding the first point, we have a dedicated internal website to operationally monitor radar system health in real time (Lorente et al., 2016). This automated quality control application analyses a variety of diagnose parameters (SNR3, etc.), considered as indicators of possible malfunctions or abnormal status, to evaluate site performance. Abrupt or gradual degradation and failure problems can be easily detected, triggering alerts for troubleshooting. The alert algorithm, based on a ternary flag system, has been implemented to detect anomalies and categorize them in order to create a historic database of flagged radial files for a later offline reprocessing when one (or more) HFR site status is (are) considered to be working abnormally. According to this tool, the radar sites overall performance and their day-to-day operation were robust and within tolerance ranges during the analysed period. Furthermore, GDOP values remained below the imposed threshold (as defined in the methodology section) in order to screen out the less reliable data.

Regarding the second point, both eulerian and lagrangian validation exercises were conducted during 2021-2022 within the frame of RADAR ON RAIA project, an Interreg España-Portugal (POCTEP) programme. HFR-derived hourly surface currents were compared against Silleiro fixed buoy (denoted in Figure 1a) and drifter buoys measurements. Results revealed a good agreement for both components (correlation in the ranges 0.53-0.74), in accordance with results previously reported in the literature (Cosoli et al., 2010; Kaplan et al., 2005). RMSE was higher for the meridional component than for the zonal one: 9.86 versus 7.65 cm·s$^{-1}$.

In addition, for the Bay of Biscay HFR it can be highlighted some of the previous work where the radar data has already been used in combination with other observations to study surface coastal transport processes and showed good agreement (Rubio et al., 2011 and 2018; Solabarrieta et al., 2016, Manso-Narvarte et al. 2018 and 2021).

**Although no additional paragraphs have been added to the manuscript to describe the quality control applied to HFR current data, we are willing to do so under request if the reviewer considers this is pertinent.**

References:

Cosoli, S., Mazzoldi, A. and Gacic, M. Validation of surface current measurements in the Northern Adriatic Sea from High Frequency radars, Journal of Atmospheric and Oceanic Technology, Vol. 27, pages 908-919, 2010.

Kaplan, D.M., Largier, J. and Botsford, L.W. HF radar observations of surface circulation off Bodega Bay (northern California, USA). Journal of Geophysical Research, Vol. 110, C10020, pages 1-25, 2005.

Lorente, P., Piedracoba S., Soto-Navarro, J., Ruiz M.I., Alvarez-Fanjul E. and Montero, P. The High Frequency coastal radar network operated by Puertos del Estado (Spain): roadmap to a fully operational implementation. IEEE Journal of Oceanic Engineering, doi: 10.1109/JOE.2016.2539438, pages 1-17, May 2016.

Manso-Narvarte, I.; Rubio, A.; Jordà, G.; Carpenter, J.; Merckelbach, L.; Caballero, A. Three-Dimensional Characterization of a Coastal Mode-Water Eddy from Multiplatform Observations and a Data Reconstruction Method. Remote Sens. 2021, 13, 674. https://doi.org/10.3390/rs13040674

Manso-Narvarte, I., Caballero, A., Rubio, A., Dufau, C., and Birol, F.: Joint analysis of coastal altimetry and high-frequency (HF) radar data: observability of seasonal and mesoscale ocean dynamics in the Bay of Biscay, Ocean Sci., 14, 1265–1281, https://doi.org/10.5194/os-14-1265-2018, 2018.

Rubio A, Reverdin G, Fontán, A., González, M., Mader, J., 2011. Mapping near-inertial variability in the SE Bay of Biscay from HF radar data and two offshore moored buoys. Geophys. Res. Lett., 38 (19): L19607.

Rubio A, Solabarrieta L, González M, Mader J, Castanedo S, Medina R, Charria G., Aranda J.A., 2013. Surface circulation and Lagrangian transport in the SE Bay of Biscay from HF radar data. OCEANS - Bergen, 2013 MTS/IEEE, doi: 10.1109/OCEANS-Bergen.2013.6608039

Rubio, A., Caballero, A., Orfila, A., Hernández-Carrasco, I., Ferrer, L., González, M., Solabarrieta, L., Mader, J., 2018. Remote Sensing of Environment, 205, 290-304, doi: 10.1016/j.rse.2017.10.037

Solabarrieta, L., Frolov, S., Cook, M., Paduan,J., Rubio, A., González,M., Mader, J., Charria, G., 2016. Skill assessment of HF radar-derived products for lagrangian simulations in the Bay of Biscay. J. Atmos. Oceanic Technol., 33, 2585–2597, doi: 10.1175/JTECH-D-16-0045.1.

6) page 9 line 280: «with» respect to
Done!

**REVIEWER-2 (anonymous)**

The manuscript by Lorente et al. investigates an alternative index for tracking the presence and strength of coastal upwelling and downwelling circulation patterns based on observed and modeled surface currents. In the case of the observed currents, analyses are presented from two coastal regions with time series extending over several months in each case. For the case of the modeled currents, an analysis is presented for one of the geographic regions for an entire 12-month period.

The surface-current-based coastal upwelling index (CUI) is compared against the traditional wind-based index. In that sense, there is no available ground-truth observation of vertical upwelling current. Independent evidence of upwelling circulation is provided in the form of sea surface temperature and chlorophyll observations whose spatial and temporal patterns match those predicted by the large events in the CUI indices.

The proposed surface-current-based CUI utilizes observations from various networks of high frequency (HF) radar installations. The availability of those observations is growing as more coastal HF radar sites are being added in many parts of the world. Extending the utility of HF radar observations is, therefore, of interest to a wide range of marine scientists and resource managers. This manuscript is generally well written and documented and the results support the use of a surface-current-based CUI. For these reasons, I recommend the manuscript for publication with only minor corrections.

**Many thanks to the anonymous reviewer-2 for the detailed review and the number of useful tips provided. Please find below a thorough point-by-point response with the hope of improving the quality of the document to make it acceptable for final publication. All those minor comments provided by the reviewer have been carefully addressed.**

Overall, the manuscript is well motivated and documented with references from the community. If anything, the Introduction could be condensed because it is slightly repetitive and long compared with the results section.

**The introduction has been shortened by 10 lines as some redundant information has been removed and few paragraphs have been reformulated and condensed.**

The main results are well documented, and they support the idea that surface-current-based CUI can be used as an alternative to an overwater wind-based CUI. I do think that the conclusions section could focus more on why a surface-current-based CUI is advantageous.

**The following paragraph has been inserted in the conclusions section to clarify the benefits of the proposed CUI-HFR:**

**"In this context, the proposed CUI-HFR presents additional advantages with respect to previous traditional CUIs, namely:**

**i) it takes into consideration the direct influence of coastal surface water dynamics, providing thereby a more complete portrait of this phenomenon.**

**ii) it provides high-resolution two-dimensional maps that can aid to elucidate the spatial distribution and magnitude of the coastal UPW together with the potential existence of recurrent patterns and/or filaments in intricate regions with complex-geometry configurations.**

**iii) it is generated from consistent remote-sensed hourly surface current observations (obtained in near real-time), not from coarse-resolution atmospheric forecasts which are in general affected by higher uncertainties. This interpretation is supported by the fact that operational atmospheric and ocean models include assimilation schemes where remote observations are routinely ingested to improve their predictive skills (Wilczak et al., 2019; Hernández-Lasheras et al., 2021)."**

**References:**

**Wilczak, JM, Olson, JB, Djalalova, I, et al. Data assimilation impact of in situ and remote sensing meteorological observations on wind power forecasts during the first Wind Forecast Improvement Project (WFIP). Wind Energy. 2019; 22: 932– 944. https://doi.org/10.1002/we.2332**

**Hernández-Lasheras, J., Mourre, B., Orfila, A., Santana, A., Reyes, E., and Tintoré, J.: Evaluating high-frequency radar data assimilation impact in coastal ocean operational modelling, Ocean Sci., 17, 1157–1175, https://doi.org/10.5194/os-17-1157-2021, 2021.**

I am a little skeptical that using a numerical circulation model to obtain surface currents to then estimate a CUI is better than simply using the winds that drive the model.

**Since coastal upwelling is a process strongly influenced by the wind but also modulated by the local bathymetry, the coastal morphology or the coastline orientation, we humbly consider that GLOBAL circulation model (with its well-known limitations) might act as a useful tool for CUI assessment as it takes into account the secondary (but not negligible) role of the abovementioned factors.**

**Although we have previously listed some advantages related to HFR-CUI, we do not intend to categorize this novel approach as better than previous traditional wind-based methodologies. All of them are valid (even complementary). In the same line of thought, in this paper we have presented a proof-of-concept investigation to assess the prognostic capabilities of the GLOBAL circulation model to accurately reproduce UPW/DOW events in the NWI area. This brief exploration might establish new pathways for future research but does not aim at ranking existing CUIs, which is out of the scope of the present paper.**

**All in all, the authors are convinced that both wind and circulation forecast models have still room for improvement. Powerful techniques such as data assimilation and machine learning will likely lead to more precise, robust predictions and therefore to more reliable CUIs. In this context, we guess that the development and operational implementation of a high-resolution fully coupled atmospheric-ocean model could constitute a step ahead to better reproduce this coastal process.**

The main benefit of a surface-current-based CUI that is suggested in the manuscript is the possibility to create a 2-D map of the CUI. For that additional level of CUI fidelity to be meaningful there should be some discussion of the relevant divergence scale that is controlling the upwelling process. The traditional wind-based CUI assumes a very large horizontal scale with surface currents diverging from the coastal boundary being responsible for the upwelling circulation. Two-dimensional observations of surface currents from HF radar (or a numerical circulation model) can, in theory, expose horizontal divergence in the flow field and the associated upwelling patterns. Such direct observations of divergence are very sensitive to errors and I'm not convinced by the results in this manuscript that the two-dimensional variations in CUI are meaningful. There should be, at a minimum, some discussion of scale and that fact that the mapping results are suggestive at best.

**The authors fully agree with the reviewer that the horizontal divergence (DIV) at the sea surface is a useful diagnostic to discriminate between zones of contraction and expansion of the flow where vertical fluxes might be significant. Indeed, we already computed maps of DIV from HF radar current observations in the same study region (Figure 5, in Lorente et al., 2020) to unveil localized areas of upwelling (UPW) and downwelling (DOW) associated with positive and negative DIV, respectively. As stated in Lorente et al. (2020), under UPW-favourable winds, positive divergence is exposed in the central portion of the radar domain and also in the periphery of Cape Finisterre, indicating accumulated upward vertical motions and strong UPW. The analysis of DIV corroborated not only the key role of the Galician shoreline orientation in modulating UPW conditions but also the importance of Cape Finisterre promontory and its ambient waters as a locus of recurrent positive DIV and offshore advection, independently of the dominant along-shore wind regime. This is in agreement with previous historical works in the same region (Torres et al., 2003; Álvarez et al., 2011; McClain et al., 1986).**

**In the present manuscript, we firstly decided not to include a discussion about the divergence scale that is controlling this coastal process in order to: i) avoid potential redundancies and overlapping with Lorente et al. (2020); and ii) fulfill the journal requirements for the "Ocean State Report" Special Issue (limit of 4 Figures). Following the reviewer's suggestion, we have added the following paragraph in the conclusions section that is further supported by those previous findings exposed in Lorente et al. (2020):**

**"The small-scale belt of UPW, confined in shallower coastal areas and evidenced in Figure 3 (a, c), is consistent with HFR-derived maps of horizontal divergence previously published in Lorente et al. (2020). In this work, it was suggested that positive divergence, localized at the tip of Cape Finisterre, induced topographic UPW and then upwelled waters were advected southwards away from the promontory. Similar initiatives with HFR current observations were effectively addressed in the west coast of the USA (Roughan et al., 2005), proposing that confined areas of semi-persistent UPW were not due to local or remote wind forcing but rather to the divergence of the prevailing southerly flow as it passed the Point Loma headland."**

**References:**

**Lorente, P.; Piedracoba, S.; Montero, P.; Sotillo, M.G.; Ruiz, M.I.; Álvarez-Fanjul, E. Comparative Analysis of Summer Upwelling and Downwelling Events in NW Spain: A Model-Observations Approach. Remote Sens. 2020, 12, 2762. https://doi.org/10.3390/rs12172762**

Torres, R.; Barton, E.D.; Miller, P.; Álvarez-Fanjul, E. Spatial patterns of wind and sea surface temperature in the Galician upwelling region. J. Geophys. Res. 2003, 108, 3130

Álvarez, I.; Gómez-Gesteira, M.; deCastro, M.; Lorenzo, M.N.; Crespo, A.J.C.; Dias, J.M. Comparative analysis of upwelling influence between the western and northern coast of the Iberian Peninsula. Cont. Shelf Res. 2011, 31, 388–399

McClain, C.R.; Chao, S.Y.; Atkinson, L.P.; Blanton, J.O.; Decastillejo, F. Wind-Driven Upwelling in the Vicinity of Cape Finisterre, Spain. J. Geophys. Res.-Ocean. 1986, 91, 8470–8486

Roughan, M.; Terril, E.J.; Largier, J.L.; Otero, M. Observations of divergence and upwelling around Point Loma, California. J. Geophys. Res. 2005, 110.

**MINOR COMMENTS:**

Line 12: "ecosystems, impacting on" should be "ecosystems, which has impacts on"

**Done!**

Line 28: "As the interface" should be "The interface"

**Done, the entire sentence has been reformulated.**

Line 45: "process denominated Ekman" should be "process referred to as Ekman"

**Done!**

Line 74: "hence two" should be "two"

**Done!**

Line 160: "those situ" should be "those in situ"

**Done!**

Line 178: "CUI-HFR which" should be "CUI-HFR, which"

**Done!**